

# Seasonal behaviour of tidal damping and residual water level slope in the Yangtze River estuary: identifying the critical position and river discharge for maximum tidal damping

Huayang Cai[1,2,5], Hubert H. G. Savenije[3], Erwan Garel[4], Xianyi Zhang[1,2],
Leicheng Guo[5], Min Zhang[6], Feng Liu[1,2], and Qingshu Yang[1,2]

[1]Institute of Estuarine and Coastal Research, School of Marine Engineering and Technology, Sun Yat-sen University, Guangzhou, China

[2]Guangdong Provincial Engineering Research Center of Coasts, Islands and Reefs, Guangzhou, China

[3]Department of Water Management, Faculty of Civil Engineering and Geosciences, Delft University of Technology, Delft, the Netherlands

[4]Centre for Marine and Environmental Research (CIMA), University of Algarve, Portugal

[5]State Key Laboratory of Estuarine and Coastal Research, East China Normal University, Shanghai, China

[6]Shanghai Normal University, Department of Geography, Shanghai, China

*Correspondence to:* Feng Liu (liuf53@mail.sysu.edu.cn)

**Abstract.** As a tide propagates into the estuary, river discharge affects tidal damping primarily through a friction term, attenuating tidal motion by increasing the quadratic velocity in the numerator, while reducing the effective friction by increasing the water depth in the denominator. For the first time, we also demonstrate a third effect of river discharge that may lead to the weakening of the

channel convergence (i.e., landward reduction of channel width and/or depth). In this study, monthly averaged tidal water levels (2003-2014) at six gauging stations along the Yangtze River estuary were used to understand the seasonal behaviour of tidal damping and residual water level slope. Observations show that there is a critical value of river discharge, beyond which the tidal damping is reduced with increasing river discharge. This phenomenon is clearly observed in the upstream part

of the Yangtze River estuary (between the Maanshan and Wuhu reach), which suggests an important cumulative effect of residual water level on tide-river dynamics. To understand the underlying mechanism, an analytical model has been used to quantify the seasonal behaviour of tide-river dynamics and the corresponding residual water level slope under various external forcing conditions. It was shown that a critical position along the estuary is where there is maximum tidal damping

(approximately corresponding to a maximum residual water level slope), upstream of which tidal damping is reduced in the landward direction. Moreover, contrary to the common assumption that larger river discharge leads to heavier damping, we demonstrate that beyond a critical value tidal damping is slightly reduced with increasing river discharge, owing to the cumulative effect of resid-





ual water level on the effective friction and channel convergence. Our contribution describes the
seasonal patterns of tide-river dynamics in detail, which will, hopefully, enhance our understanding
of the nonlinear tide-river interplay and guide effective and sustainable water management in the
Yangtze River estuary and other estuaries with substantial freshwater discharge.

## 1   Introduction

Tide-river interactions and resulting residual water level profiles play a crucial role in large-scale
river deltas (e.g., the Mississippi River delta in the United States, the Rhine-Meuse delta in the
Netherlands, the Pearl River delta and the Yangtze River delta in China, the Ganges-Brahmaputra
delta in Bangladesh, etc.) because tide-river dynamics exert a tremendous impact on delta mor-
phodynamics, salt intrusion, and deltaic ecosystems (Hoitink and Jay, 2016; Hoitink et al., 2017).
However, it is only in recent years that substantial effort has been devoted to the nonlinear interaction
between tidal waves and riverine flow in estuaries (e.g., Kukulka and Jay, 2003a; Buschman et al.,
2009; Lamb et al., 2012; Sassi and Hoitink, 2013; Guo et al., 2014, 2015, 2016; Cai et al., 2014a,b,
2016, 2018; Leonardi et al., 2015). Hence, many aspects of tide-river interactions (e.g., seasonal
behaviour of tidal damping and residual water level slope) deserve further exploration.

The impact of river discharge on tidal wave propagation, especially on tidal damping, in estuaries
has long been the subject of intensive scientific interest (e.g., Dronkers, 1964; LeBlond, 1979; Godin,
1985, 1999; Jay, 1991; Horrevoets et al., 2004; Kukulka and Jay, 2003b; Cai et al., 2012a, 2014b,
2016; Guo et al., 2015; Leonardi et al., 2015; Alebregtse and de Swart, 2016; Zhang et al., 2018).
It is worth noting that traditional methods for analysing tidal signals (e.g., harmonic and Fourier
analysis) are restricted due to the assumption of stationary signals. To correctly understand tidal
wave behaviour under the influence of river discharge, non-stationary tidal harmonic analysis has
been developed to better account for the nonlinear tide-river interactions (e.g., Jay and Flinchem,
1997, 1999; Kukulka and Jay, 2003a; Jay et al., 2011, 2015; Matte et al., 2013, 2014). Generally, it
was shown that river discharge tends to attenuate tidal energy and hence to enhance tidal damping
primarily through bottom friction (e.g., Godin, 1985, 1999; Guo et al., 2015). Recently, building on
a variety of previous studies on tidal damping (e.g., Horrevoets et al., 2004; Savenije et al., 2008; Cai
et al., 2012a,b; Savenije, 2012), Cai et al. (2014b, 2016) proposed an analytical hydrodynamic model
to investigate the underlying mechanism of tide-river interaction by means of an envelope method,
where an analytical expression for tidal damping can be obtained by subtracting high water and low
water envelopes. It is important to note that the river discharge impacts tidal damping primarily
via the friction term in the momentum equation: on the one hand, attenuating the tidal motion via
increasing the quadratic velocity in the numerator; and on the other hand, reducing the effective
friction by increasing the residual water level (hence water depth) in the denominator. This effect is
well illustrated in extreme cases where dredging along the upper estuary has significantly increased



the mean water depth resulting in strong tidal amplification for a given discharge (e.g., Jay et al.,
2011). However, little effort has been devoted to exploring the effect of river discharge on channel
convergence, which is the other control factor for tide-river dynamics.

Although the important role played by the residual water level on estuarine hydrodynamics was
recognized for some time (e.g., LeBlond, 1979; Godin and Martinez, 1994), only a few studies
explored the effects of a dynamic residual water level slope on tide-river dynamics (e.g., Buschman
et al., 2009; Sassi and Hoitink, 2013; Cai et al., 2014b, 2016). It is well-known that the steady
gradient of residual water level is mainly induced by a residual frictional effect (e.g., Cai et al.,
2014b, 2016), a density effect (e.g., Savenije, 2005, 2012) and nonlinear advective acceleration.
However, it should be noted that the effects of density and advective acceleration are generally
minor when compared with the frictional effect (this is detailed more in section 3.1). In addition,
the nonlinear tide-river interactions can be linearized by decomposing the friction term into different
components contributed by tidal forcing, river flow, and tide-river interaction alone (e.g., Buschman
et al., 2009; Sassi and Hoitink, 2013; Cai et al., 2016). This was made by using the Chebyshev
polynomials approach to approximate the quadratic velocity in the friction term (Dronkers, 1964;
Godin, 1991, 1999). In general, in the tide-dominated reach the residual water level is primarily
determined by tide–river interaction, whereas it is mainly controlled by the river flow alone in the
river-dominated reach (Cai et al., 2016).

The tide-river dynamics in the Yangtze River estuary, located on the east coast of China, have re-
ceived increasing attention in recent years owning to intensive climate change and human interven-
tions (e.g., Three Gorges Dam construction, Deep Waterway Project) on both riverine and marine
processes (e.g., Cai et al., 2014b,a, 2016; Guo et al., 2015; Zhang et al., 2015a,b; Alebregtse and
de Swart, 2016; Kuang et al., 2017; Shi et al., 2018; Zhang et al., 2018). Traditionally, a time-series
analysis method (such as harmonic analysis in a non-stationary mode and continuous wavelet trans-
forms) was adopted to identify the non-stationary tide-river behaviour along the estuary axis based
on observed data or results from numerical models, such that the impacts of river discharge on differ-
ent tidal constituents can be quantified separately (e.g., Guo et al., 2015; Zhang et al., 2015a,b; Shi
et al., 2018; Zhang et al., 2018). Recently, idealized (or analytical) models with a strongly simplified
geometry and flow characteristics were applied to the Yangtze River estuary in order to reproduce the
first-order features of tide-river dynamics (e.g., Cai et al., 2014b,a, 2016; Alebregtse and de Swart,
2016). It is important to note that the idealized model proposed by Alebregtse and de Swart (2016)
adopted a uniform depth for each channel, thus neglecting the residual water level caused by the
strong tide-river interaction. As a result, their model is only applicable to the lower region of the
Yangtze River estuary, where the tide dominates the river flow. In contrast, the analytical model pro-
posed by Cai et al. (2014b, 2016) accounting for the effects of residual water level can reasonably
reproduce the first-order tide-river dynamics (only considering a predominant tidal constituent, e.g.,
$M_2$) in the Yangtze River estuary for a wide range of tide and river discharge conditions. Although





many studies have been undertaken to understand the tide-river interactions in the Yangtze River estuary, previous studies mainly focused on the tidal properties near the estuary mouth and investigations of tide-river dynamics are limited for the whole estuary, especially in the transitional zone with strongly nonlinear tide-river interactions. In this study, we adopt the analytical model proposed

by Cai et al. (2014b, 2016) to quantify the impacts of river discharge on the seasonal behaviour of tide-river dynamics (e.g., tidal damping) and residual water level slope.

The remainder of this paper is constructed as follows. An overview of the study area and datasets used to study the seasonal behaviour of tidal damping and residual water level slope are described in section 2. Section 3 introduces the analytical hydrodynamic model for reproducing tide-river

dynamics in estuaries. The main results illustrating the seasonal behaviour of tidal damping and residual water level slope are presented in section 4, after which a discussion is presented in section 5. Finally, some conclusions are drawn in section 6.

## 2   Overview of the Yangtze River estuary

### 2.1   Description of the study site

The Yangtze River estuary, located on the east coast of China, extends ∼630 km from the Datong hydrological station (where the tidal limit is) to its mouth near the seaward end of the South Branch (Figure 1). Both tidal waves and river flow are the major sources of energy for the hydrodynamics along the Yangtze River estuary. Specifically, the estuary is a meso-tidal type with a maximum and mean tidal range of 4.62 and 2.67 m near the estuary mouth, respectively. The tide has an

irregular semidiurnal character with average flood and ebb duration of 5 and 7.5 hr, respectively (Zhang et al., 2012). According to the observed data in the Datong hydrological station (1950-2012), the annual mean river discharge is approximately 28,200 m³/s and the monthly average river discharge reaches a maximum value of 49,500 m³/s in July and a minimum value of 11,300 m³/s in January. Unlike previous studies focusing on the tidal hydrodynamics near the estuary mouth, here

we mainly concentrate on the tide-river dynamics in the mainstream of the Yangtze River estuary, extending from Tianshenggang gauging station to the Datong hydrological station.

### 2.2   Datasets

Monthly averaged hydrological data (including tidal range and water level) from 6 tidal gauging stations (Tianshenggang: TSG; Jiangyin: JY, located 46 km upstream of TSG; Zhenjiang: ZJ, located

155 km upstream of TSG; Nanjing: NJ, located 236 km upstream of TSG; Maanshan: MAS, located 284 km upstream of TSG; Wuhu: WH, located 330 km upstream of TSG) along the Yangtze River estuary were collected from the Yangtze Hydrology Bureau of the People's Republic of China for the period of 2003-2014. The tidal amplitude is determined by averaging the flood and ebb tidal amplitudes. To correctly calculate the residual water level slope, measured water levels from the





gauging stations have been corrected to the Huanghai 1985 datum of local mean sea level. Figure 2 illustrates the temporal variation of the tidal amplitude and water level observed at the 6 gauging stations together with the monthly averaged river discharge observed at the Datong (DT) hydrological station. In Figure 2, we observe a strongly seasonal variation in tidal range $H$ (except for TSG and JY) and residual water level $\overline{Z}$ due to the strongly fluctuating river discharge. For the residual water

level, we also note that the more upstream the location of the station, the more evident the seasonal change is.

## 3   Analytical model for tide-river dynamics

### 3.1   Reproducing the residual water level profile in estuaries

The dynamics of residual water level can be derived from the one-dimensional momentum equation

(e.g., Savenije, 2005, 2012):

$$\frac{\partial U}{\partial t} + U \frac{\partial U}{\partial x} + g \frac{\partial Z}{\partial x} + \frac{gh}{2\rho} \frac{\partial \rho}{\partial x} + g \frac{U|U|}{K^2 h^{4/3}} = 0, \tag{1}$$

where $U$ is the cross-sectional averaged velocity, $Z$ is free surface elevation, $h$ is water depth, $g$ is the acceleration of gravity, $t$ is time, $\rho$ is water density, $x$ is the longitudinal coordinate directed landward, and $K$ is the Manning-Strickler friction coefficient. Assuming a periodic variation of flow

velocity, the integration of Equation (1) over a tidal cycle leads to an expression for the residual water level slope (e.g., Vignoli et al., 2003; Cai et al., 2014b, 2016):

$$\frac{\partial \overline{Z}}{\partial x} = -\frac{1}{K^2} \overline{\left( \frac{U|U|}{h^{4/3}} \right)} - \frac{1}{2g} \frac{\partial \overline{U^2}}{\partial x} - \frac{1}{2\rho_0} h \overline{\frac{\partial \rho}{\partial x}}, \tag{2}$$

where the over bars and the subscript 0 indicate the tidal average and the value at the seaward boundary, respectively. As shown in Equation (2) the residual water level slope is caused by three

contributions made by residual friction, advective acceleration and density effects that correspond to the three terms on the right-hand side of Equation (2). Note that the contribution from advective acceleration to the residual water level slope:

$$\frac{\partial \overline{Z}_{adv}}{\partial x} = -\frac{1}{2g} \frac{\partial \overline{U^2}}{\partial x}, \tag{3}$$

can be easily integrated to:

$$\overline{Z}_{adv} = -\frac{1}{2g} \left( \overline{U^2} - \overline{U_0^2} \right) = -\frac{1}{2} \overline{Fr_0} \left( \frac{\overline{U^2}}{\overline{U_0^2}} - 1 \right) \overline{h_0}, \tag{4}$$





where we introduced the Froude number, $\overline{Fr^2} = \overline{U^2}/(g\overline{h})$, computed with the averaged variables. In this case, the correction is local (not cumulative) and proportional to the flow depth through a coefficient that is negligible as long as the velocity does not change significantly, and $Fr$ is small, as is common for most tidal flows. With regard to the contribution from the density effect, it was shown by Savenije (2005, 2012) that the induced value of residual water level only amounts to around 1.25% of the estuary depth over the salt intrusion length. Hence, in this study we neglect the impact of density on the dynamics of residual water level.

Assuming negligible advective acceleration influence and density effect, integration of Equation (2) leads to an expression for the residual water level:

$$\overline{Z}(x) = -\int_0^x \overline{\frac{\partial Z}{\partial x}} = -\int_0^x \overline{\frac{U|U|}{K^2 h^{4/3}}}, \tag{5}$$

if we assume $\overline{Z} = 0$ at the estuary mouth (where $x=0$).

### 3.2 Analytical solution for tide-river dynamics

To correctly reproduce the residual water level profile in estuaries, an iterative procedure is required to properly calculate the friction term as presented in Equation (5) because the velocity amplitude and water depth are unknown a priori. This is made by using the analytical hydrodynamic model proposed by Cai et al. (2016). In the analytical model, the fundamental assumption made for the geometry of the estuary is that the tidally averaged cross-sectional area $\overline{A}$ and width $\overline{B}$ can be approximated by the exponential functions (e.g., Cai et al., 2016):

$$\overline{A} = \overline{A_r} + (\overline{A_0} - \overline{A_r})\exp(-x/a), \tag{6}$$

$$\overline{B} = \overline{B_r} + (\overline{B_0} - \overline{B_r})\exp(-x/b), \tag{7}$$

where $\overline{A_0}$ and $\overline{B_0}$ are the cross-sectional area and width at the estuary mouth, $\overline{A_r}$ and $\overline{B_r}$ represent the asymptotic riverine cross-sectional area and width, whereas $a$ and $b$ denote the convergence lengths of the cross-sectional area and width, respectively. The tidally averaged depth $\overline{h}$ can be computed directly following the assumption of a mostly rectangular cross-section, i.e., $\overline{h} = \overline{A}/\overline{B}$. The possible impact of the lateral storage areas (e.g., tidal flats or salt marshes) can be described by the storage width ratio $r_S = B_S/\overline{B}$ defined as the ratio of the storage width $B_S$ and the tidally averaged stream width $\overline{B}$.

Concentrating on a predominantly tidal constituent (e.g., $M_2$) with frequency $\omega$, it was shown by Cai et al. (2016) that tide-river dynamics are mainly determined by four externally defined, dimensionless parameters (see Table 1), i.e., the dimensionless tidal amplitude $\zeta$ defined as the ratio of the





tidal amplitude to the depth, the estuary shape number $\gamma$ (describing the cross-sectional area convergence), the friction number $\chi$ (representing the role of frictional dissipation), and the dimensionless river discharge $\varphi$ (indicating the influence of freshwater discharge $Q$ imposed at the upstream boundary), where $\eta$ is the tidal amplitude, $\upsilon$ is the velocity amplitude, $U_r = Q/\overline{A}$ is the river flow velocity and $c_0$ is the classical wave celerity in a prismatic frictionless channel, defined as $c_0 = \sqrt{g\overline{h}/r_S}$.

The analytical solutions for the main tide-river dynamics can be obtained by solving a set of four implicit equations:

the damping/amplification equation:

$$\delta = \frac{\mu^2 \left(\gamma\theta - \chi\mu\lambda\Gamma\right)}{1 + \mu^2\beta}, \tag{8}$$

the phase lag equation:

$$\tan\left(\varepsilon\right) = \frac{\lambda}{\gamma - \delta}, \tag{9}$$

the scaling equation:

$$\mu = \frac{\sin(\varepsilon)}{\lambda} = \frac{\cos(\varepsilon)}{\gamma - \delta}, \tag{10}$$

the celerity equation:

$$\lambda = 1 - \delta(\gamma - \delta). \tag{11}$$

The main dependent dimensionless parameters in Equations (8)-(11) are presented in Table 1. They include the amplification/damping number $\delta$, describing the rate of increase ($\delta > 0$), or decrease ($\delta < 0$) of the tidal wave amplitude along the channel axis, the velocity number $\mu$ indicating the ratio of the actual velocity amplitude to the reference value in a prismatic frictionless channel, $\lambda$ the celerity number denoting the ratio of the classical wave celerity $c_0$ to the actual wave celerity $c$, and $\varepsilon$ the phase lag between high water (HW) and high water slack (HWS) or between low water (LW) and low water slack (LWS), where $\phi_Z$ and $\phi_U$ are the water level phase and velocity, respectively. The unknown parameters $\theta$, $\beta$, and $\Gamma$ in the damping/amplification equation account for the impact of river discharge, where $\theta$ and $\beta$ are defined in Table 1, and $\Gamma$ is given by:

$$\Gamma = \frac{1}{\pi} \left[ p_1 - 2p_2\varphi + p_3\varphi^2 \left(3 + \mu^2\lambda^2/\varphi^2\right) \right], \tag{12}$$

which is a friction factor derived by means of a Chebyshev polynomials approach (Cai et al., 2016). In Equation (12), $p_i$ ($i$=0, 1, 2, 3) are the Chebyschev coefficients (see Dronkers, 1964, p.301), which are functions of the dimensionless river discharge $\varphi$ through $\alpha = \arccos(-\varphi)$:





$$p_0 = -\frac{7}{120}\sin\left(2\alpha\right) + \frac{1}{24}\sin\left(6\alpha\right) - \frac{1}{60}\sin\left(8\alpha\right), \tag{13}$$

$$p_1 = \frac{7}{6}\sin\left(\alpha\right) - \frac{7}{30}\left(3\alpha\right) - \frac{7}{30}\sin\left(5\alpha\right) + \frac{1}{10}\sin\left(7\alpha\right), \tag{14}$$

$$p_2 = \pi - 2\alpha + \frac{1}{3}\sin\left(2\alpha\right) + \frac{19}{30}\sin\left(4\alpha\right) - \frac{1}{5}\sin\left(6\alpha\right), \tag{15}$$

$$p_3 = \frac{4}{3}\sin\left(\alpha\right) - \frac{2}{3}\sin\left(3\alpha\right) + \frac{2}{15}\sin\left(5\alpha\right). \tag{16}$$

The set of Equations (8)-(11) can be regarded as a consistent analytical framework for understanding tide-river dynamics in estuaries. For more details about the computation procedure, readers can refer to Cai et al. (2014a,b, 2016).

## 4   Results

### 4.1   Seasonal variation in tidal damping rate and residual water level slope

To understand the importance of seasonal changes in river discharge on tide-river dynamics, we first explore the seasonal variation of the tidal damping rate and the residual water level slope (Figure 3). Here, the tidal damping rate $\delta_H$ and the residual water level slope $S$ an be estimated for a reach of length $\Delta x$:

$$\delta_H = \frac{1}{\left(\eta_1 + \eta_2\right)/2}\frac{\eta_2 - \eta_1}{\Delta x}, \tag{17}$$

$$S = \frac{\overline{Z_2} - \overline{Z_1}}{\Delta x}, \tag{18}$$

where $\eta_1$ and $\overline{Z_1}$ are the tidal amplitude and residual water level on the seaward side, respectively, whereas $\eta_2$ and $\overline{Z_2}$ are the corresponding values at a distance $\Delta x$ upstream, respectively.

The study period covers tide-river dynamics under both low and high flow conditions, where the monthly average river discharge observed at the DT station ranges from approximately 9,174 to 61,400 m$^3$/s so that the nonlinear interaction between tidal wave and river flow varies considerably. It has been previously shown that river discharge impacts the tidal damping rate and residual water level slope primarily through the friction term (Cai et al., 2014b, 2016). It can be clearly seen





from Figure 3 that both the tidal damping rate and residual water level slope vary strongly with river discharge. Remarkably, it appears there is a threshold, corresponding to a critical value of river discharge, beyond which the relationship between the tidal damping rate and river discharge switches from negatively to positively correlated (Figure 3a). This is particularly the case in the upper reach between the MAS and WH stations when the river discharge threshold is approximately 35,000 m$^3$/s. In Figure 3b, it appears that the residual water level slope is linearly correlated with river discharge.

### 4.2 Performance of the analytical model

The main geometric characteristics (including the tidally averaged cross-sectional area, width and depth) used in this paper were extracted from a digital elevation model (DEM) produced from Yangtze River estuary navigation charts surveyed in 2007. The elevations have been corrected to the local mean sea level of the Huanghai1985 datum. Figure 4 displays the longitudinal geometric quantities along the Yangtze River estuary axis, in combination with the best-fitting curves reproduced by functions (6) and (7). Table 2 shows the calibrated geometric characteristics, where we observe a relatively large value of cross-sectional area convergence length (151 km), with a relatively small value for width (44 km), suggesting a fast transition from a funnel shaped reach to a prismatic reach in terms of width.

The analytical model was calibrated and verified against the observed tidal amplitude and residual water level along the Yangtze River estuary on the basis of the monthly averaged hydrological data during 2003-2014. The adopted seaward tidal amplitude (at the TSG station) and upward river discharge (at the DT station) in the analytical model are displayed in Figure 2. Since the Yangtze River estuary features a typical semidiurnal character, for the sake of simplification, we used a typical M$_2$ tidal period (i.e., 12.42 hr). The only calibrated parameter in the analytical model is the Manning-Strickler friction coefficient $K$. The storage width ratio $r_S$ was assumed as $r_S$ =1. The calibrated value of $K$ is 80 m$^{1/3}$s$^{-1}$ in the seaward region ($x$=0-42 km), whereas a smaller value of $K$=55 m$^{1/3}$s$^{-1}$ is used in the river dominated region ($x$=42-450 km). Figure 5 shows a comparison between the observed and computed tidal amplitude and residual water level at different gauging stations along the Yangtze River estuary for a wide range of tide and river discharge conditions. We observe that the correspondence between analytically computed results and observations is good with a high value for the coefficient of determination ($R^2 > 0.96$), suggesting a reasonable performance of the analytical model for reproducing the first-order tide-river dynamics along the Yangtze River estuary. However, we note an overestimation of the analytically computed residual water level at upstream stations (i.e., MAS and WH) for values >5 m, which is likely due to the oversimplification of the geometry and flow characteristics (e.g., neglecting the M$_4$ and M$_6$ overtides) in the analytical model.



### 4.3 Seasonal behaviour of tide-river dynamics

The calibrated analytical model is subsequently used to explore the response of the main tide-river dynamics (represented by the damping/amplification number $\delta$, the velocity number $\mu$, the celerity number $\lambda$ and the phase lag $\varepsilon$) to the seasonal variation of river discharge (Figure 6 and Figures S1-S3). Figure 6a shows the spatial-temporal patterns of the damping number $\delta$ for the studied period (2003-2014), together with its minimum value $\delta_{min}$ indicating the maximum tidal damping. The most noticeable feature of the spatial-temporal pattern of tidal damping is the seasonal variation with river discharge, which is clearly illustrated by the temporal variation of the tidal damping critical value $\delta_{min}$ (see red line in Figure 6a, varying between 233 and 500 km in the Yangtze estuary). To be more specific, in Figure 6b, it can be observed that the critical value of tidal damping and its position along the estuary are negatively correlated with the corresponding river discharge. Generally, the critical value of tidal damping $\delta_{min}$ reaches its minimum in December or January when the monthly average river discharge is minimum, and reaches its maximum in July with maximum river discharge throughout the year. Similarly, we observe that the position along the estuary with the critical $\delta_{min}$ reaches its maximum value for minimum river discharge, and minimum value for maximum river discharge. Similar seasonal behaviour of velocity amplitude (denoted by the velocity number $\mu$), wave celerity (denoted by the celerity number $\lambda$) and phase lag (denoted by $\varepsilon$) can also be reproduced using the calibrated analytical model (see Figures S1-S3 in the Supplemental Material). In general, we observe a negatively correlated relation between $\mu$, $\varepsilon$ and $Q$, and a positively correlated relation between $\lambda$ and $Q$. In addition, we note that the seasonal behaviour of the critical phase lag (i.e., minimum value) is relatively irregular (see Figure S3 in the Supplemental Material) since it is determined by the changes in $\gamma$, $\lambda$, and $\delta$, following the phase lag equation $\tan(\varepsilon) = \lambda/(\gamma - \delta)$ (see Equation 9).

### 4.4 Seasonal behaviour of the residual water level slope

For a typical tidal river, it is usually observed that the tidal range is reduced when the residual water level rises in the landward direction owing to the residual water level slope, which is mainly balanced by the residual frictional effect (Cai et al., 2014b, 2016). To understand the underlying mechanism of tidal wave propagation under the influence of river discharge, we adopted a decomposition method that can be used to quantify the contributions of tide, river and tide-river interaction to the residual water level slope $S$ (see detailed derivation in Cai et al., 2016), computed as:

$$S = -\overline{\frac{1}{K^2 \overline{h}^{4/3} \pi} (p_0 \upsilon^2 + p_1 \upsilon U + p_2 U^2 + p_3 U^3/\upsilon)}, \tag{19}$$

Equation (19) can be decomposed into three components contributing to the increase of residual water level:

a tidal component:





$$S_t = \frac{1}{K^2 \overline{h}^{4/3} \pi} \left( \frac{1}{2} p_2 + p_0 \right) v^2, \tag{20}$$

a riverine component:

$$S_r = \frac{1}{K^2 \overline{h}^{4/3} \pi} \left( p_2 - p_3 \varphi \right) U_r^2, \tag{21}$$

and tide-river interaction component:

$$S_{tr} = \frac{1}{K^2 \overline{h}^{4/3} \pi} \left( -p_1 - \frac{3}{2} p_3 \right) v U_r. \tag{22}$$

Figure 7 shows the seasonal variation of the residual water level slope $S$, exhibiting a positively
correlated relationship with the river discharge. It can be seen from Figure 7a that the temporal
variation of critical value $S_{max}$ (maximum value) is quite similar to the tidal damping (see Figure
6a), which suggests that the development of tide-river dynamics (e.g., tidal damping) is closely
related to the residual water level slope. To further understand the relative importance of tidal forcing
alone, river flow alone and tide-river interactions, Equations (20)-(22) were used to quantify the
contributions made by both tidal and riverine forcing. The results of these separate components are
presented in Figure S4 (see the Supplemental Material), where we observe that the main contribution
lies in the riverine component $S_r$.

## 5   Discussion

### 5.1   Critical position of maximum tidal damping (corresponding to the minimum value of
damping number $\delta$)

To understand the main processes that control the development of a maximum tidal damping, we
used the average values of the observed tidal amplitude at the TSG station and the river discharge
at the DT station as model inputs and reproduced the main tide-river dynamics along the Yangtze
estuary. Figure 8 shows the longitudinal variation of the main tidal dynamics ($\delta$, $\lambda$, $\mu$, and $\varepsilon$) and
the contributions made by both tidal and riverine forcing to the residual water level slope together
with the water depth for both the wet (Figures 8a, c, e) and dry seasons (Figures 8b, d, f). The
discontinuous jump occurring at $x$=42 km depends on the adoption of different friction coefficients
in the analytical model. Apparently, the critical position of maximum tidal damping is closer to the
sea side during wet season (around $x$=305 km) than the dry season (around $x$=410 km) owing to
the river discharge flush. In addition, the position of maximum tidal damping (corresponding to the
minimum value of damping number $\delta$, indicated by the dashed black line) is almost coincident with
the minimum values of the wave celerity $\lambda$ and velocity $\mu$ numbers. The slightly lagged responses of



$\lambda$ and $\mu$ to $\delta$ are due to nonlinear interaction between these main tide-river dynamics parameters, as described by the set of nonlinear Equations (8)-(11). The change in phase lag $\varepsilon$ is directly followed

by the phase lag equation $\tan(\varepsilon) = \lambda/(\gamma - \delta)$ (see Equation 9). The underlying mechanism generating the maximum tidal damping can be clearly shown in Figures 8e, and f, where we observe that the residual water level slope $S$ and its dominant river component $S_r$ increase to a maximum value near the critical position of maximum tidal damping, beyond which it is reduced. This means that the main tide-river dynamics are driven by the residual water level slope and that the critical position

of maximum tidal damping is primarily controlled by the riverine forcing component. Furthermore, we also note that the maximum value of $S$ corresponds to the local minimum $\overline{h}$, which suggests a dominant impact of residual water level (hence water depth) on tide-river dynamics in the Yangtze River estuary.

It is also worth examining the longitudinal and seasonal variations of the two controlling parame-

ters represented by the estuary shape number $\gamma$ and the friction number $\chi$ (see Figure 9), which are closely related to the strength of tidal damping $\delta$. Remarkably, it is important to note that the effect of channel convergence (represented by $\gamma$) is stronger during the dry season (larger value of $\gamma$) than wet season. This indicates that river discharge also tends to reduce the channel convergence through generation of the residual water level slope. In addition, we observe a switch of $\gamma$ from positive

to negative at $x$=290 km and $x$=394 km for the wet and dry seasons, respectively (Figure 9a). The cause of the negative value for $\gamma$ is that the cross-sectional area increases in the landward direction (hence $\mathrm{d}\overline{A}/\mathrm{d}x > 0$) owing to the increasing residual water level and depth. In Figure 9b, we observe a larger value for the friction number $\chi$ during the dry season than wet season, which is mainly due to the relatively larger tidal amplitude during the dry season and the residual water level (hence the

water depth) increasing with river discharge (see the definition of $\chi$ in Table 1). Furthermore, it is also noted that $\chi$ asymptotically approaches 0 with distance. This means that in the upstream part of the estuary tide-river dynamics are primarily determined by the geometric effect (i.e., the divergence of the cross-sectional area) and the residual frictional effect caused by riverine forcing $S_t$ (see Equation 21).

## 5.2  Critical river discharge for maximum tidal damping

Based on the analytical results, in Figure 10 we display how the tidal damping $\delta$, the residual water level slope Sand the residual water level $\overline{Z}$ develop as a function of river discharge $Q$ for different positions in the upstream river-dominated region, where the maximum tidal damping occurs. Figure 10a shows the tidal damping at different positions with different river discharges. It also displays the

critical value of river discharge corresponding to maximum tidal damping. As expected, more river discharge is required to change tidal damping from a negative gradient (indicating a strengthening damping) to a positive gradient (indicating a weakening damping) for the seaward positions where tide exerts more influence. The critical river discharge is approximately 34,000 $\mathrm{m^3s^{-1}}$ at $x$=470 km,





and it gradually increases to 55,000 m$^3$s$^{-1}$ at $x$=350 km. In Figure 10a, we also note that beyond
the critical value of maximum damping the $\delta$ appears to slightly increase until an asymptotic value
is approached. Figures 10b and 10c show the relation between $S$, $\overline{Z}$ and $Q$. It is noticeable that
the curves for residual water level $\overline{Z}$ appear as straight lines, corresponding to a consistent increase
of residual water level slope $S$ with river discharge $Q$. Unlike the longitudinal variation of the
maximum $S$ value (see Figures 8e, f), both $S$ and $\overline{Z}$ monotonously increase with $Q$.

The underlying mechanism for achieving a critical river discharge for maximum tidal damping can
be primarily attributed to the cumulative effect of residual water level $\overline{Z}$ altering the water depth and
hence the channel convergence and effective friction, according to the definitions of estuary shape
number $\gamma$ and friction number $\chi$ in Table 1. Figure 11 presents these two controlling parameters
($\gamma$ and $\chi$) as a function of river discharge $Q$. It can be clearly seen in Figure 11a that there exists
an apparent switch of the estuary shape number $\gamma$ from positive (indicating a reduction of cross-
sectional area in the landward direction) to negative (indicating an increase of cross-sectional area in
the landward direction). In addition, more river discharge is required to achieve a switch in estuary
shape number $\gamma$ for the seaward positions where tide exerts more influence. The main reason for
such a switch is the consistent increase of residual water level and hence water depth and cross-
sectional area with river discharge. On the other hand, the effective friction induced by tidal forcing
(represented by $\chi$) asymptotically approaches 0 with the river discharge (see Figure 11b), which
suggests that the estuarine system is primarily controlled by the divergence of the cross-sectional
area and the residual frictional effect caused by riverine forcing (represented by $S_r$ in Equation 21)
for high river discharge conditions. Additionally, we can conclude that the asymptotic behaviour
of tidal damping $\delta$ with high river discharge (as shown in Figure 10a) is due to the corresponding
asymptotic behaviour of estuary shape number $\gamma$ and friction number $\chi$ (and hence the residual
frictional effect indicated by $S$ as presented in Figure 10b).

### 5.3 Implications for sustainable water management and sediment transport

Knowledge of the development and evolution of tide-river dynamics that determine the behaviour of
tidal damping and residual water level slope under external forcing (e.g., tidal and riverine flow) and
geometry changes (e.g., deepening and land reclamation) are essential for improving the sustain-
able water management in estuaries. Adopting the method proposed in this study, one can evaluate
the influence of human interventions occurred in the estuarine system (such as large-scale sand ex-
cavation, dredging for navigational channel or freshwater withdrawal) on flood control structures
(e.g. storm surge barriers, flood gates), and aquatic environment (e.g., such as salt intrusion and
the related water quality). When combined with ecological or salt intrusion models, the analytical
approach presented in this study is particularly useful for a quick computation of the longitudinal
distribution of the salinity (e.g., Cai et al., 2015). Using salinity as a general predictor, it is possible
to assess the potential impacts of human interventions on the aquatic ecosystem health in general



(e.g., water quality, water utilization and agricultural development in the estuarine area).

As tide propagates into an estuary, it is distorted and becomes asymmetric due to significant nonlinear interactions with geometry and river flow. Tidal asymmetry is regarded as one of the most important mechanisms generating residual sediment transport (e.g., Friedrichs and Aubrey, 1988; Parker, 1991; Guo et al., 2014, 2015, 2016). Although the current analytical method can
only deal with tide-river interaction for a single predominant tidal constituent (e.g., $M_2$), the model does capture the major tidal asymmetry induced by geometric effect and riverine flow (e.g., the tidal asymmetry caused by the residual river flow) and can well reproduce the seasonal behaviour of tidal damping and residual water level slope. It was shown by Lamb et al. (2012) that the erosion and deposition patterns along an estuary are strongly related to the shape of the residual water level
profile, which we have shown to be linked to the tide-river dynamics and the geometry of the estuary. The successful reproduction of the seasonal behaviour of tide-river dynamics and residual water level slope in the Yangtze estuary suggests that the proposed analytical approach can be used as a tool for detecting the evolution of estuarine morphology under various external forcing conditions. However, further studies are required to quantify the relationship between the residual water level slope and
the estuarine morphology.

## 6   Conclusions

Both observations and analytical model results show a critical value of river discharge that causes maximum tidal damping in the upstream part of the tidal river, challenging the concept of how river discharge dampens tidal waves. The residual water level slope, mainly balanced by the residual
frictional effect, plays a key role in determining the evolution of tide-river dynamics under a wide range of tidal and riverine forcing conditions. A critical position along the estuary is where there is maximum tidal damping, upstream of which the residual water level slope is reduced. The location of this position moves seaward with the increase of river discharge. From that position landwards, the effect of river discharge on tidal damping becomes weaker instead of stronger, indicating a
weakening of the backwater effect induced by the residual frictional effect.

Moreover, analytical model results show that more river discharge is required to change the maximum tidal damping critical value from a negative to a positive gradient for the seaward positions where the tide exerts stronger impact. The underlying mechanism has to do with the fact that river discharge affects tidal damping, on the one hand, attenuating tidal motion by increasing the quadratic
velocity in the numerator, and on the other hand, reducing the effective friction by increasing the water depth in the denominator. The occurrence of critical river discharge suggests the cumulative effect of residual water level (increasing the water depth and the cross-sectional area) that exceeds the threshold of tide-river dynamics, beyond which tidal damping weakens with river discharge. To the best of our knowledge, this is the first study that shows the gradient switch of the cross-sectional



area (i.e., $\mathrm{d}\overline{A}/\mathrm{d}x$) and tidal damping (i.e., $\mathrm{d}\delta/\mathrm{d}x$) with the river discharge, shedding new light on the impact of river discharge on tidal damping in alluvial estuaries. Moreover, the results obtained in this study will, hopefully, provide scientific guidelines for water resources management (e.g., flood control and salt intrusion prevention) in the Yangtze River estuary and other tidal rivers worldwide.

*Acknowledgements.* We acknowledge the financial support from the Open Research Fund of State Key Laboratory of Estuarine and Coastal Research (Grant No. SKLEC-KF201809), from the National Natural Science Foundation of China (Grant No. 51709287, 41106015, 41476073, 41506105, 41876091), from the Basic Research Program of Sun Yat-Sen University (Grant No. 17lgzd12), and from the Guangdong Provincial Natural Science Foundation of China (Grant No. 2017A030310321). The work of Erwan Garel was supported by FCT research contract IF/00661/2014/CP1234.





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





**Table 1.** Dimensionless parameters adopted in the analytical model for tide-river dynamics

| Local variables | Dependent variables |
|---|---|
| Dimensionless tidal amplitude | Amplification number |
| $\zeta = \eta/\overline{h}$ | $\delta = c_0\,\mathrm{d}\eta/(\eta\omega\mathrm{d}x)$ |
| Estuary shape number | Velocity number |
| $\gamma = c_0(\overline{A} - \overline{A_r})/(\omega a\overline{A})$ | $\mu = \upsilon/(r_{\mathrm{S}}\zeta c_0) = \upsilon\overline{h}/(r_{\mathrm{S}}\eta c_0)$ |
| Friction number | Celerity number |
| $\chi = r_{\mathrm{S}}gc_0\zeta\left[1 - (4\zeta/3)^2\right]^{-1}/(\omega K^2\overline{h})$ | $\lambda = c_0/c$ |
| Dimensionless River discharge | Phase lag |
| $\varphi = U_{\mathrm{r}}/\upsilon$ | $\varepsilon = \pi/2 - (\phi_Z - \phi_U)$ |
| $\beta = \theta - r_{\mathrm{S}}\zeta\varphi/(\mu\lambda), \quad \theta = 1 - (\sqrt{1+\zeta} - 1)\varphi/(\mu\lambda)$ | |

**Table 2.** Characteristics of geometric parameters in the Yangtze River estuary

| Characteristics | River | Mouth | Convergence length $a$ or $b$ (km) |
|---|---|---|---|
| Cross-sectional area $\overline{A}$ (m$^2$) | 12,135 | 51,776 | 151 |
| Width $\overline{B}$ (m) | 2005 | 6735 | 44 |

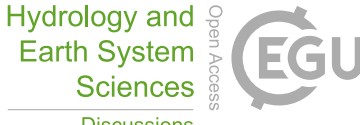


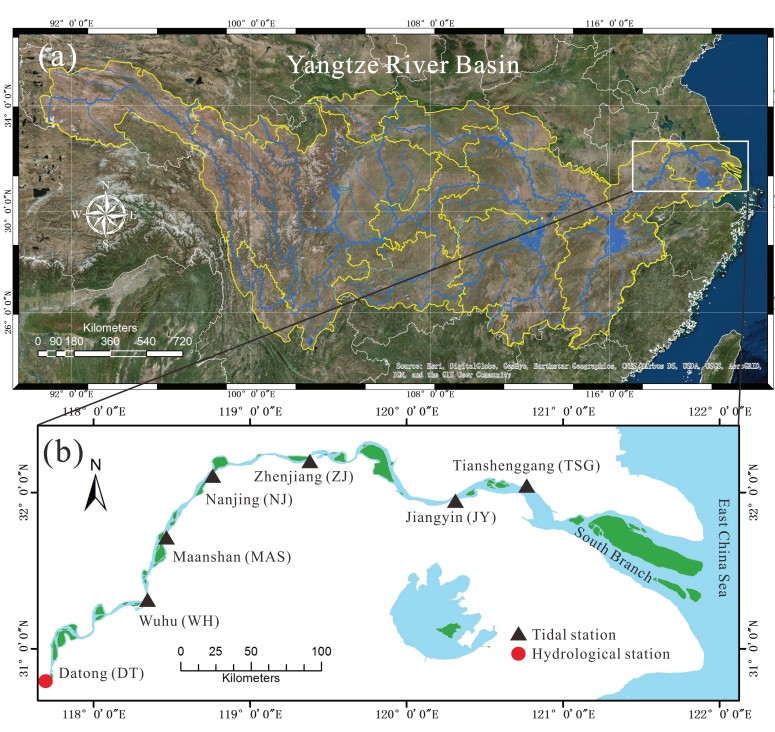

**Fig. 1.** Sketch map of the Yangtze River basin (a) and the Yangtze River estuary (b) displaying the location of gauging (triangle) and hydrological (circle) stations.





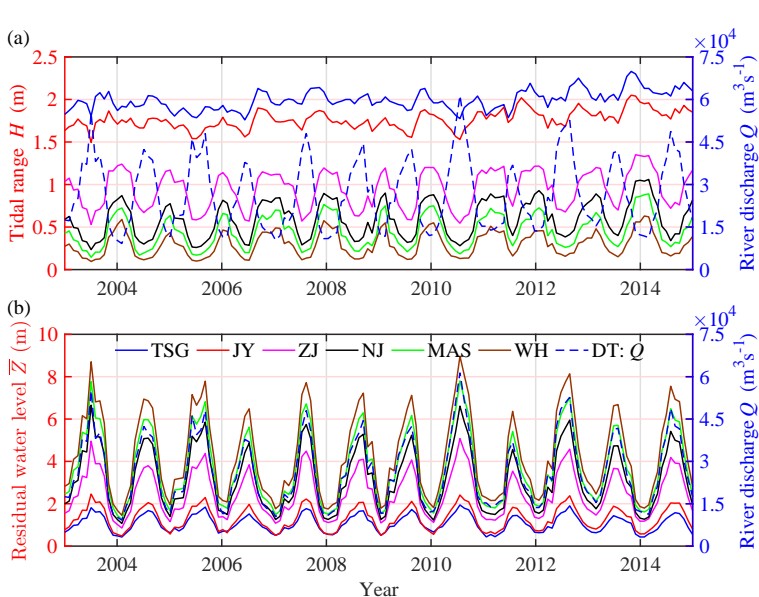

**Fig. 2.** Temporal (monthly averaged) variations of observed tidal range $H$ (a) and residual water level $\overline{Z}$ (b) at different gauging stations along the Yangtze River estuary together with the observed river discharge at Datong hydrological station.





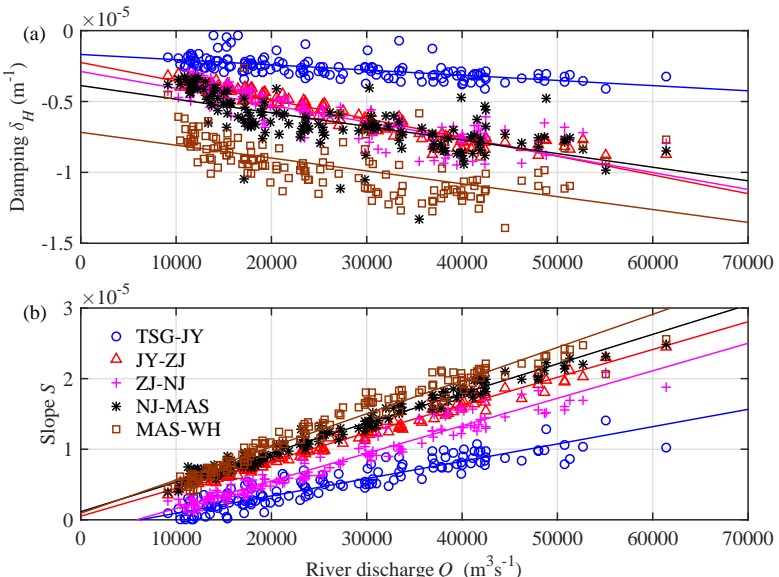

**Fig. 3.** Scatterplot and liner regression line of tidal damping rate $\delta_H$ (a) and residual water level slope $S$ (b) for different reaches in the Yangtze River estuary as a function of river discharge observed at the DT hydrological station.

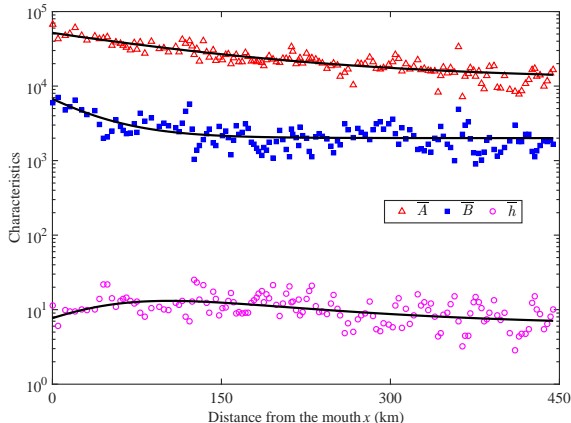

**Fig. 4.** Longitudinal variation of the main geometric characteristics (cross-sectional area, width and depth) along the Yangtze River estuary. The thick black lines represent the best fitting curves.





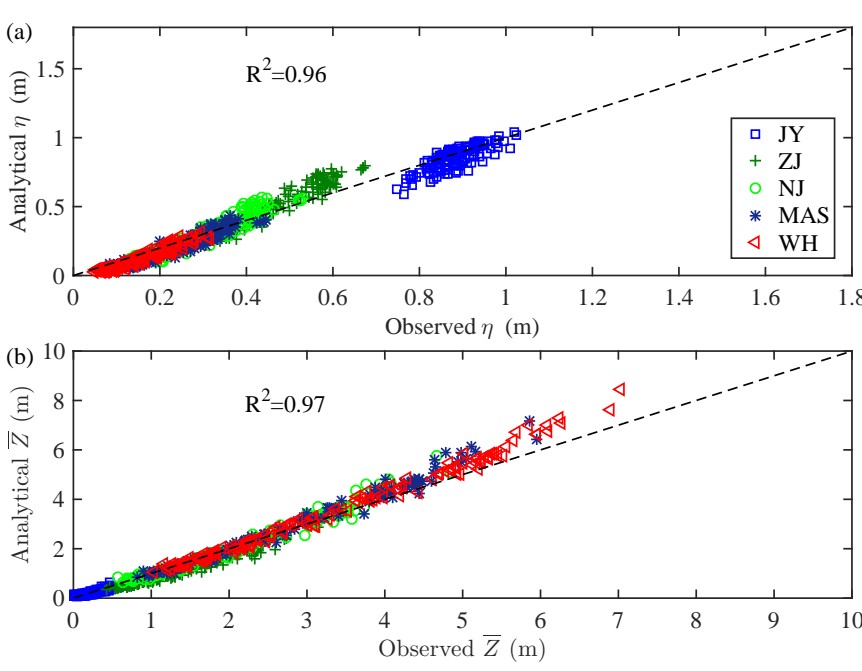

**Fig. 5.** Comparison of analytically computed tidal amplitude $\eta$ (a) and residual water level $\overline{Z}$ (b) against the observations in the Yangtze River estuary during the study period (2003-2014).





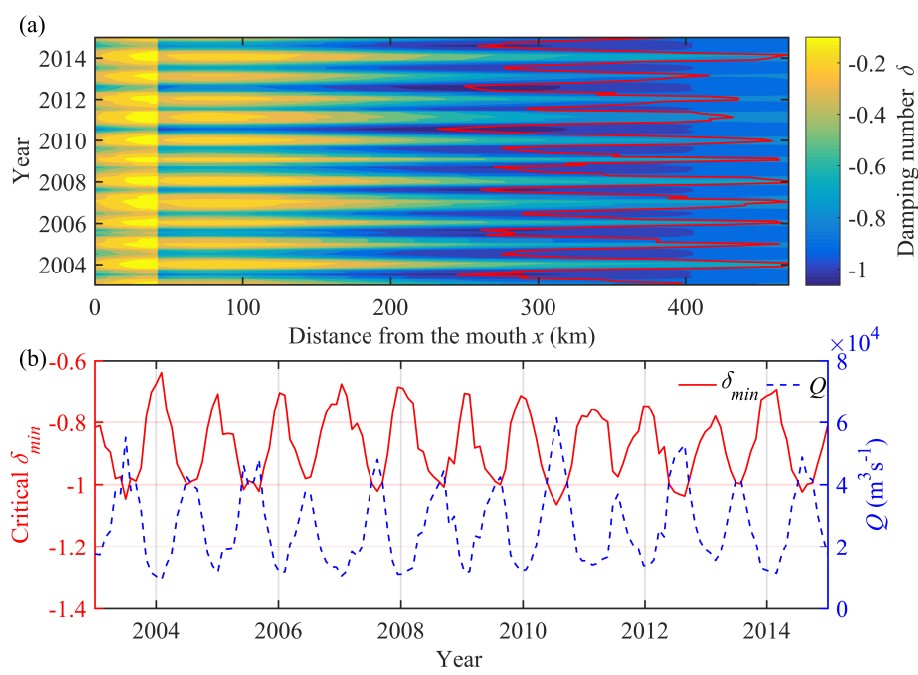

**Fig. 6.** Contour plot of the damping number $\delta$ together with its minimum value $\delta_{min}$ (indicated by the red line) for each month (a) and the relation between the critical value and river discharge $Q$ (b).

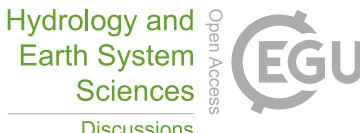



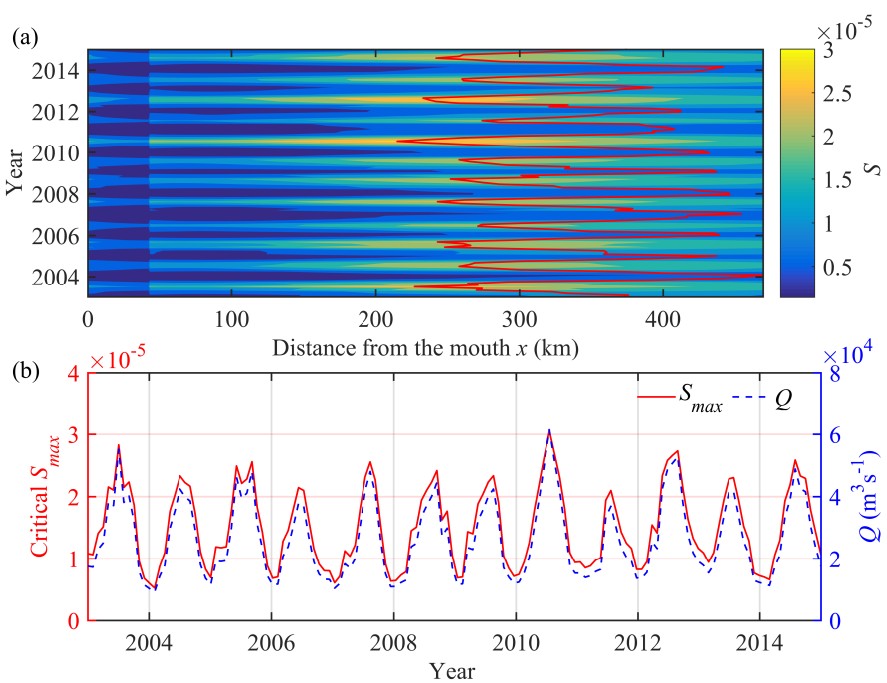

**Fig. 7.** Contour plot of the residual water level slope $S$ together with its minimum value $S_{max}$ (indicated by the red line) for each month (a) and the relation between the critical value and river discharge $Q$ (b).





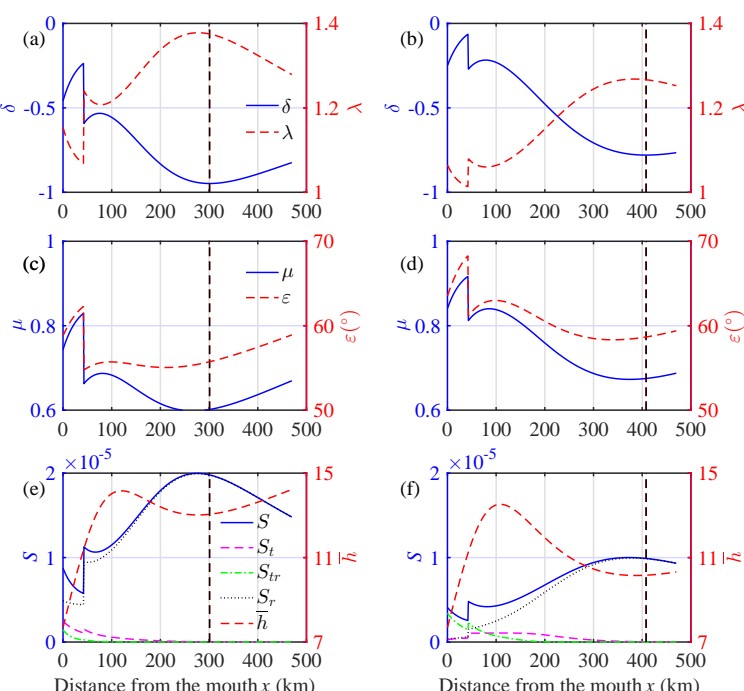

**Fig. 8.** Longitudinal variation of the main tide-river dynamics (a, b, c, d) and contributions of tidal and riverine forcing to the residual water level slope together with the water depth (e, f) for the wet (a, c, e) and dry seasons (b, d, f) in the Yangtze estuary. The dashed lines in each subplot represent the critical position for maximum tidal damping (corresponding to the minimum value of damping number $\delta$).

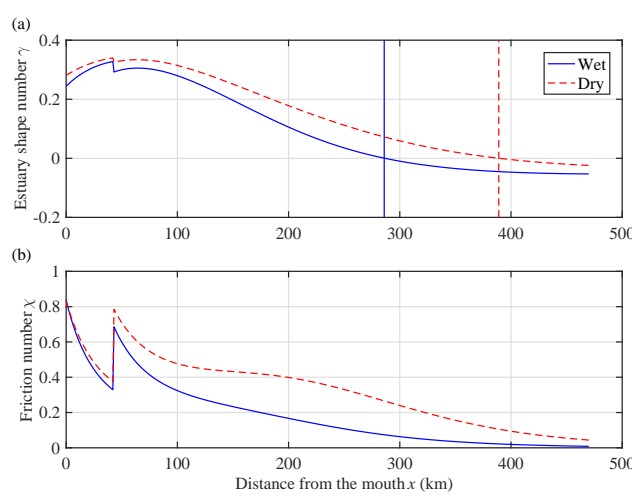

**Fig. 9.** Longitudinal variation of the estuary shape number $\gamma$ (a) and the friction number $\chi$ (b) for the wet and dry seasons in the Yangtze estuary. Subplot (a) also indicates the position of the critical value of channel convergence (i.e., $\gamma=0$) using the corresponding lines for the wet and dry seasons.





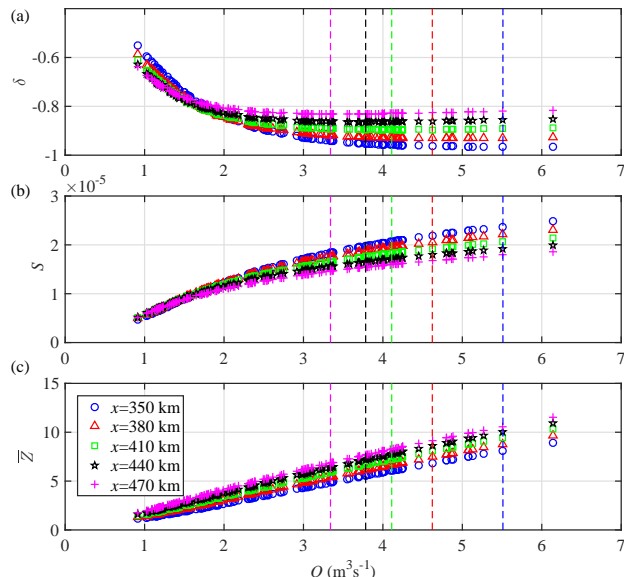

**Fig. 10.** Relationship between the tidal damping number $\delta$ (a), the residual water level slope $S$ (b), the residual water level $\overline{Z}$ (c) and the corresponding river discharge $Q$ imposed at the DT hydrological station for different positions, indicated by different symbols. The dashed lines with the same colour as the symbols were used to identify the critical river discharge for the maximum tidal damping (corresponding to the minimum value of $\delta$ in subplot a).

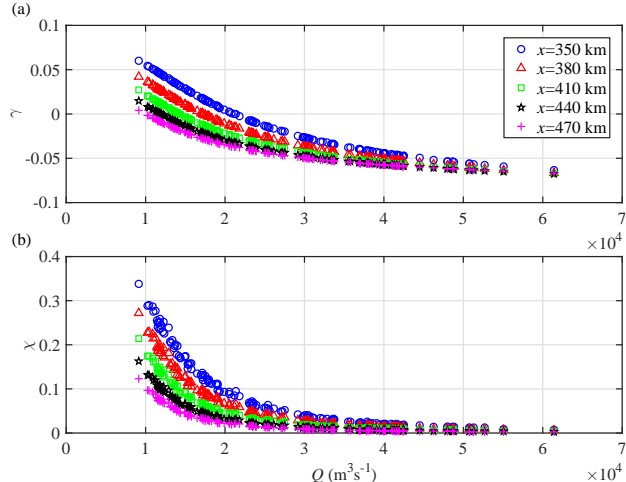

**Fig. 11.** Relationship between the estuary shape number $\gamma$ (a), the friction number $\chi$ (b) and the river discharge $Q$.