# Peer review of "Seasonal behaviour of tidal damping and residual water level slope in the Yangtze River estuary: identifying the critical position and river discharge for maximum tidal damping"

_Hydrology and Earth System Sciences, 2018_

## Referee Comment (RC1) · Anonymous Referee #1 · 16 Jan 2019

This manuscript investigates the influence of river discharge on tidal damping and residual water level slopes in the Yangtze River estuary at the seasonal scale. Building on previous work by the same author(s), an analytical model for river-tide dynamics is used to understand the underlying mechanisms responsible for the observed variability. Of particular interest, the authors identified (1) a critical value of river discharge, at a given location, beyond which tidal damping is reduced with increasing discharge, and (2) a critical position along the estuary, for a given discharge (e.g. wet or dry season), upstream of which tidal damping is reduced in the landward direction. Although the

methods used were presented before, this application to a large estuary reveals new insights into the seasonal patterns of river-tide dynamics, which have implications for sustainable water management and sediment transport, as stressed by the authors. The subject is thus quite relevant; the manuscript is well written, well documented and clearly structured. The result analysis is thorough and a good discussion is presented. Overall, this is a very good paper. I recommend its publication after minor revision. My comments are detailed below.

General comments:

Do the critical values found in (1) and (2) (see above) represent the same phenomenon after all? It seems like, for a given (constant) discharge, when you move upstream a tidal river, the relative influence of river discharge increases, which is analogous to a discharge increase at a given (fixed) location. Is this a good reasoning? If relevant, a word on the similarity/dissimilarity between the two processes could be said.

Discussion: A word on applicability/transferability of the method to other systems or other dynamical contexts should be added in the discussion. In particular, would this analytical approach work in systems with mixed diurnal/semidiurnal tides, in non-convergent estuaries, or in estuaries with irregular (non-rectangular) cross-sections? Would it be possible to reduce the temporal averaging window to analyse the neap-spring variability in tidal damping and residual water level slopes? Similarly, could the method be adapted to rapidly varying flows? What adaptations would be necessary to include these aspects, if possible? I am not asking that the authors make those changes, but a discussion on limitations (and possible upgrades) of the proposed methodology would be useful.

The Yangtze River estuary does not seem to have sharp morphological breaks, based on Fig. 4. However, in systems where they occur, a shift in the tidal-fluvial conditions may be observed near these breaks. In such a case, the location of the boundary between the tide-dominated and river-dominated reaches may be invariant to changes

in river flow (Hoitink & Jay, 2016). In this situation, what should be expected to be the consequence on the position of maximum tidal damping and maximum residual water level slope along the estuary, under different discharge conditions?

Specific comments:

L55-56: "the effect of river discharge on channel convergence, which is the other control factor for tide-river dynamics": Can you provide a reference, or is this new knowledge? Also, it is not quite clear at this stage in the manuscript how discharge can affect channel convergence. Can you explain in a few words?

L81-90: "Recently, idealized (or analytical) models with a strongly simplified geometry and flow characteristics were applied [...]. [The model] can reasonably reproduce the first-order tide-river dynamics (only considering a predominant tidal constituent)": In terms of justification of the method, considering the limitations of the analytical model, can you explain the interest or benefit of using such a simplified model compared to full numerical models?

L92: "previous studies mainly focused on the tidal properties near the estuary mouth": Can you please provide references supporting this affirmation?

L123-124: "The tidal amplitude is determined by averaging the flood and ebb tidal amplitudes": What do you mean by "flood and ebb tidal amplitudes"? Please clarify.

L126: Consider adding "monthly averaged" before "tidal amplitude and water level" and replacing "tidal amplitude" by "tidal range", if appropriate.

L232-235 and Fig. 3a: "there is a threshold, corresponding to a critical value of river discharge, beyond which the relationship between the tidal damping rate and river discharge switches from negatively to positively correlated": This does not show clearly in Fig. 3a, because of the straight regression lines. Can you illustrate the observed shifts with dotted lines maybe?

L255: Using two different friction coefficients K for the seaward and landward regions

creates a break in the results. Is it real? If not, consider making a smoother transition of friction between the two regions.

L284-285 and Fig. S3: "the seasonal behaviour of the critical phase lag is relatively irregular": It looks regular to me in Fig. S3, but inversely correlated with Q. Please adjust the text accordingly.

L304-308: It should be said more explicitly that both the maximum value of residual water level slope $S_{max}$ and its position along the estuary (Fig. 7) are correlated with river discharge.

L310-312 and Fig. S4: Can you briefly explain why the position of $S_{r,max}$ is landward of the other two terms ($S_{t,max}$ and $S_{tr,max}$)?

L326-327: The position of maximum tidal damping is almost coincident with the maximum (not minimum) values of the wave celerity and the minimum values of the velocity number. Please correct.

L327-328: "The slightly lagged responses [. . .] are due to nonlinear interaction between these main tide-river dynamics parameters": Are you able to provide a more detailed physical explanation for it?

L329-330: Replace "is directly followed by" by "directly follows from". Can you explain the correlation between the phase lag $\varepsilon$ and the other variables and its role in tidal wave propagation (damping and celerity) based on your results?

L353: Replace $S_t$ by $S_r$.

L361-362: Please specify whether the negative (positive) gradient indicates a strengthening (weakening) damping with respect to Q or to the landward position along the estuary, or both.

L434-435: "this is the first study that shows the gradient switch of the cross-sectional area and tidal damping with the river discharge": This gradient switch in tidal damping

was also recently documented by Matte et al., (2019) in the St. Lawrence River at the neap-spring and seasonal scale. I suggest referencing their work, here or elsewhere in the manuscript.

L451-456: There is a duplication of references: Cai et al. (2012a) and Cai et al. (2012b) are the same.

Fig. 1: Can you add river kilometers at each station in panel (b)?

Figs. 2 and 8: I find it hard to differentiate the pink, red and/or dark red curves. Can you use more contrasting colors?

Fig. 10: Add "x 10ˆ4" to the scale for river discharge Q. In Fig. 10a, there is another gradient switch happening around 15 000 $m^3$/s with respect to the position along the estuary. At lower discharges, maximum damping occurs seaward, whereas at higher discharges, it occurs landward. This was not described in the text (section 5.2), although explanations are provided elsewhere in the manuscript. Still, it might be worth pointing out the occurrence of this other gradient switch in Fig. 10a and relating it with the gradient switch that occurs with the increasing discharge (appearing in the same figure).

References:

Hoitink, A. J. F., & Jay, D. A. (2016). Tidal river dynamics: Implications for deltas. Reviews of Geophysics, n/a-n/a. https://doi.org/10.1002/2015rg000507

Matte, P., Secretan, Y., & Morin, J. (2019). Drivers of residual and tidal flow variability in the St. Lawrence fluvial estuary: Influence on tidal wave propagation. Continental Shelf Research. https://doi.org/https://doi.org/10.1016/j.csr.2018.12.008

---

## Author Comment (AC1) · 20 Jan 2019

**Responses to comments by Reviewer #1**

We thank the Reviewer's careful consideration of our work. In this rebuttal, we have addressed all the comments formulated by the Reviewer by replying (in black) to her/his remarks (in blue).

This manuscript investigates the influence of river discharge on tidal damping and residual water level slopes in the Yangtze River estuary at the seasonal scale. Building on previous work by the same author(s), an analytical model for river-tide dynamics is used to understand the underlying mechanisms responsible for the observed variability. Of particular interest, the authors identified (1) a critical value of river discharge, at a given location, beyond which tidal damping is reduced with increasing discharge, and (2) a critical position along the estuary, for a given discharge (e.g. wet or dry season), upstream of which tidal damping is reduced in the landward direction. Although the methods used were presented before, this application to a large estuary reveals new insights into the seasonal patterns of river-tide dynamics, which have implications for sustainable water management and sediment transport, as stressed by the authors. The subject is thus quite relevant; the manuscript is well written, well documented and clearly structured. The result analysis is thorough and a good discussion is presented. Overall, this is a very good paper. I recommend its publication after minor revision. My comments are detailed below.

**Our reply:** We thank the Reviewer for her/his overall positive assessment of our work.

General comments:
Do the critical values found in (1) and (2) (see above) represent the same phenomenon after all? It seems like, for a given (constant) discharge, when you move upstream a tidal river, the relative influence of river discharge increases, which is analogous to a discharge increase at a given (fixed) location. Is this a good reasoning? If relevant, a word on the similarity/dissimilarity between the two processes could be said.

**Our reply:** We thank the Reviewer for this comment. Indeed, the underlying mechanism of generating critical position and river discharge is the same. In the revised paper, we shall include the following sentence in the Conclusions part:

"*It is worth noting that the underlying mechanism of generating critical position along the estuary is similar to that of generating critical river discharge due to the fact that for a given (constant) river discharge, the more upstream in a tidal river, the stronger effect caused by the river discharge, which is analogous to a river discharge increase at a given (fixed) location.*"

Discussion: A word on applicability/transferability of the method to other systems or other dynamical contexts should be added in the discussion. In particular, would this analytical approach work in systems with mixed diurnal/semidiurnal tides, in nonconvergent estuaries, or in estuaries with irregular (non-rectangular) cross-sections? Would it be possible to reduce the temporal averaging window to analyse the neap-spring variability in tidal damping and residual water level slopes? Similarly, could the method be adapted to rapidly varying flows? What adaptations would be necessary to include these aspects, if possible? I am not asking that the authors make those changes, but a discussion on limitations (and possible upgrades) of the proposed methodology would be useful.

**Our reply:** We agree with the Reviewer for this comment. In the revised paper, we shall supplement a new subsection in the Discussion part to highlight the limitation and transferability of the analytical model:

"*Although the current analytical model can well reproduce the first-order tide-river dynamics, it also has some limitations. The fundamental assumption is that the tidal wave can be described by a combination of a steady residual term (generated by the river discharge) and a time-dependent harmonic wave (introduced by the tidal flow). Thus, the proposed model can only capture the tidal asymmetry caused by tide-river interaction while it neglects the tidal asymmetry introduced by astronomical tides (e.g., nonlinear interactions among $K_1$, $O_1$ and $M_2$), overtides (e.g., $M_4$) and compound tides (e.g., MSf). Consequently, the proposed analytical method is preferably applied to tidal rivers with a predominant tidal constituent (e.g., $M_2$ or $K_1$).*

*It is assumed that both the tidally averaged cross-sectional area and channel width can be approximated by exponential functions following Equations (6)-(7). However, this is not a restrictive assumption since the model in principle can be applied to an arbitrary estuarine shape (i.e., bed elevation and channel width), as long as the variation of the cross-section is gradual. The proposed model can also be used to quantify the spring-neap variability of the tide-river dynamics based on daily averaged tidal amplitude and river discharge conditions (see example in Cai et al., 2016). However, the model cannot be used to explore the tide-river dynamics within a tidal cycle since it is based on a tidally averaged scale. This means that it may not be applicable to the cases with rapidly varying river discharge.*"

The Yangtze River estuary does not seem to have sharp morphological breaks, based on Fig. 4. However, in systems where they occur, a shift in the tidal-fluvial conditions may be observed near these breaks. In such a case, the location of the boundary between the tide-dominated and river-dominated reaches may be invariant to changes in river flow (Hoitink & Jay, 2016). In this situation, what should be expected to be the consequence on the position of maximum tidal damping and maximum residual water level slope along the estuary, under different discharge conditions?

**Our reply:** Thanks a lot for raising such an interesting case study, although the current analytical model is not applicable for such a case since it is assumed that the variation of the cross-section is gradual. In this case, it is likely to have a local maximum tidal damping at the boundary between the tide-dominated and river-dominated reaches owing to the sudden change of cross-section. However, it is possible to apply the analytical model proposed in this study to the upstream river-dominated region for given a suitable tidal forcing condition at the boundary (i.e., the downstream end of the river-dominated reach). Thus, the dynamics of position of maximum tidal damping and maximum residual water level slope along the estuary under a wide range of river discharge conditions would be the same as presented in this study. Further study in estuaries with sharp morphological breaks is needed in order to understand the underlying mechanism of tide-river dynamics.

Specific comments:

L55-56: "the effect of river discharge on channel convergence, which is the other control factor for tide-river dynamics": Can you provide a reference, or is this new knowledge? Also, it is not quite clear at this stage in the manuscript how discharge can affect channel convergence. Can you explain in a few words?

**Our reply:** In the revised paper, we shall revise the sentence into:

"*However, little effort has been devoted to exploring the effect of river discharge on channel convergence (represented by the gradient of cross-sectional area), which is the other control factor for tide-river dynamics (e.g., Matte et al., 2018, 2019). In particular, the river discharge affects the channel convergence primarily through residual water level (and hence water depth).*"

L81-90: "Recently, idealized (or analytical) models with a strongly simplified geometry and flow characteristics were applied [: : :]. [The model] can reasonably reproduce the first-order tide-river dynamics (only considering a predominant tidal constituent)": In terms of justification of the method, considering the limitations of the analytical model,

can you explain the interest or benefit of using such a simplified model compared to full numerical models?

**Our reply:** In the revised paper, we shall include the following sentences to clarify the benefit of using a simplified analytical model when compared with the full numerical model:

"*Although the tide-river dynamics in terms of elevation and velocity fields can be accurately simulated using fully nonlinear numerical models (e.g., Zhang et al., 2018), the cause–effect relations (e.g. the impact of river discharge on tidal damping) cannot be explicitly identified by single realizations of numerical runs. To this aim, analytical models are valuable instruments that can provide a straightforward insight. Additionally, analytical models only need a minimum amount of data, and can explicitly provide estimates of integral quantities (e.g. tidal amplitude, velocity amplitude, wave celerity and phase lag), while numerical models need to reconstruct them from temporal and spatial time series.*"

L92: "previous studies mainly focused on the tidal properties near the estuary mouth": Can you please provide references supporting this affirmation?

**Our reply:** In the revised paper, we shall add the following three publications:

Alebregtse, N. C. and de Swart, H. E.: Effect of river discharge and geometry on tides and net water transport in an estuarine network, an idealized model applied to the Yangtze Estuary, Cont. Shelf. Res., 123, 29–49, doi:10.1016/j.csr.2016.003.028, 2016.

Lu, S., Tong,C., Lee, D.Y., Zheng, J., Shen, J., Zhang, W., and Yan, Y.:
Propagation of tidal waves up in Yangtze Estuary during the dry season, J. Geophys. Res., 120, 6445–6473, doi:10.1002/2014JC010414, 2015.

Zhang, W., Feng, H. C., Hoitink, A. J. F., Zhu, Y. L., Gong, F.: Tidal impacts on the subtidal flow division at the main bifurcation in the Yangtze River Delta, Estuar. Coast Shelf S., 196, 301-314, doi:10.1016/j.ecss.2017.07.008, 2017.

L123-124: "The tidal amplitude is determined by averaging the flood and ebb tidal amplitudes": What do you mean by "flood and ebb tidal amplitudes"? Please clarify.

**Our reply:** In the revised paper, we shall clarify the definition of the tidal amplitude used in this paper:

"*The tidal amplitude is defined as a half of the tidal range either during the flood or the ebb period and we determined the mean value by averaging the tidal amplitudes during flood and ebb periods.*"

L126: Consider adding "monthly averaged" before "tidal amplitude and water level" and replacing "tidal amplitude" by "tidal range", if appropriate.

**Our reply:** Thanks a lot for pointing this out! We have replaced "tidal amplitude and water level" with "monthly averaged tidal range and water level".

L232-235 and Fig. 3a: "there is a threshold, corresponding to a critical value of river discharge, beyond which the relationship between the tidal damping rate and river discharge switches from negatively to positively correlated": This does not show clearly in Fig. 3a, because of the straight regression lines. Can you illustrate the observed shifts with dotted lines maybe?

**Our reply:** In order to highlight the threshold of river discharge, we shall fit the tidal damping rate using the quadratic equation. The revised figure 3 is presented below (Figure R1):

[Figure]

Figure R1. Scatterplot of tidal damping rate $\delta_H$ (a) and residual water level slope $S$ (b) for different reaches in the Yangtze River estuary as a function of river discharge observed at the DT hydrological station. Subplot (a) also presents the quadratic regression lines, while subplot (b) presents the linear regression lines.

L255: Using two different friction coefficients K for the seaward and landward regions creates a break in the results. Is it real? If not, consider making a smoother transition of friction between the two regions.

**Our reply:** We very much appreciate reviewer's comment, which is indeed helpful to improve the performance of the analytical model to reproduce the main tide-river dynamics along the estuary. In the revised paper, we shall explicitly mention that:

"*The calibrated value of K is 80 $m^{1/3}s^{-1}$ in the seaward region (x=0-32 km), whereas a smaller value of K=80 $m^{1/3}s^{-1}$ is used in the river dominated region (x=52-450 km). Meanwhile, in order to avoid discontinuous jump caused by the adoption of different friction coefficients, we adopted a friction coefficient of 80-55 $m^{1/3}s^{-1}$ (indicating a linear reduction of the friction coefficient) over the transitional reach (x=32-52 km).*"

L284-285 and Fig. S3: "the seasonal behaviour of the critical phase lag is relatively irregular": It looks regular to me in Fig. S3, but inversely correlated with Q. Please adjust the text accordingly.

**Our reply:** We thank the Reviewer for this comment. In the revised paper, we shall remove this sentence.

L304-308: It should be said more explicitly that both the maximum value of residual water level slope Smax and its position along the estuary (Fig. 7) are correlated with river discharge.

**Our reply:** In the revised paper, we have explicitly mention that:

"*This indicates that both the maximum value of residual water slope $S_{max}$ (Figure 7b) and its position along the estuary (Figure 7a) are positively correlated with river discharge.*"

L310-312 and Fig. S4: Can you briefly explain why the position of Sr,max is landward of the other two terms (St,max and Str,max)?

**Our reply:** In the revised paper, we shall explicitly mention that:

"*In addition, we note that the position of the maximum riverine component $S_r$ is landward of the corresponding maximum values of the other two contributions ($S_t$ and $S_{tr}$), which is mainly due to the relatively larger residual frictional effect introduced by the riverine forcing.*"

L326-327: The position of maximum tidal damping is almost coincident with the maximum (not minimum) values of the wave celerity and the minimum values of the velocity number. Please correct.

**Our reply:** You are right! In the revised paper, we shall correct this mistake:

"*In addition, the position of maximum tidal damping (corresponding to the minimum value of damping number δ, indicated by the dashed black line) is almost coincident with the maximum value of the celerity number λ and the minimum value of the velocity number μ.*"

L327-328: "The slightly lagged responses [: : :] are due to nonlinear interaction between these main tide-river dynamics parameters": Are you able to provide a more detailed physical explanation for it?

**Our reply:** We thank the Reviewer for this comment. In the revised paper, we shall supplement the following sentence:

"*This also indicates the significantly nonlinear effect caused by estuary shape, bottom friction and river discharge as the tidal wave propagating upriver.*"

L329-330: Replace "is directly followed by" by "directly follows from". Can you explain the correlation between the phase lag ε and the other variables and its role in tidal wave propagation (damping and celerity) based on your results?

**Our reply:** Many thanks for the correction. In the revised paper, we shall provide more details regarding the relationship between the phase lag $\varepsilon$ and the other variables:

"*As can be seen from Figures 8a-d, in general the phase lag $\varepsilon$ is positively correlated with the damping number $\delta$ and the velocity number $\mu$, while it is negatively correlated with the celerity number $\lambda$. Unlike tide-dominated estuaries with negligible residual water level, the key parameter that determines the nonlinear relationship between the phase lag $\varepsilon$ and the other variables ($\delta$, $\mu$, $\lambda$) in tidal rivers lies in the water depth, which is controlled by the dynamics of residual water level.*"

L353: Replace St by Sr.

**Our reply:** Corrected as suggested.

L361-362: Please specify whether the negative (positive) gradient indicates a strengthening (weakening) damping with respect to Q or to the landward position along the estuary, or both.

**Our reply:** Here, the negative (positive) gradient indicates a strengthening (weakening) damping with respect to river discharge $Q$. In the revised paper, we shall explicitly mention this point.

L434-435: "this is the first study that shows the gradient switch of the cross-sectional area and tidal damping with the river discharge": This gradient switch in tidal damping was also recently documented by Matte et al., (2019) in the St. Lawrence River at the neap-spring and seasonal scale. I suggest referencing their work, here or elsewhere in the manuscript.

**Our reply:** We agree with your comment. In the revised paper, we shall include two recent publications by Matte et al. (2018, 2019).

L451-456: There is a duplication of references: Cai et al. (2012a) and Cai et al. (2012b) are the same.

**Our reply:** We shall correct this mistake in the revised paper.

Fig. 1: Can you add river kilometers at each station in panel (b)?

**Our reply:** Yes! In the revised paper, we shall add river kilometers at each station in panel (b). The updated figure is presented below (Figure R2).

[Figure]

Figure R2. Sketch map of the Yangtze River basin (a) and the Yangtze River estuary (b) displaying the location of gauging (triangle) and hydrological (circle) stations.

Figs. 2 and 8: I find it hard to differentiate the pink, red and/or dark red curves. Can you use more contrasting colors?

**Our reply:** In the revised paper, we shall revise these Figures (see Figures R3 and R4 below).

[Figure]

Figure R3. Temporal (monthly averaged) variations of observed tidal range $H$ (a) and residual water level $\bar{Z}$ (b) at different gauging stations along the Yangtze River estuary together with the observed river discharge at Datong hydrological station.

[Figure]

Figure R4. Longitudinal variation of the main tide-river dynamics (a, b, c, d) and contributions of tidal and riverine forcing to the residual water level slope together with the water depth (e, f) for the wet (a, c, e) and dry seasons (b, d, f) in the Yangtze estuary. The dashed lines in each subplot represent the critical position for maximum tidal damping (corresponding to the minimum value of damping number $\delta$).

Fig. 10: Add "x 10^4" to the scale for river discharge Q. In Fig. 10a, there is another gradient switch happening around 15 000 m3/s with respect to the position along the estuary. At lower discharges, maximum damping occurs seaward, whereas at higher discharges, it occurs landward. This was not described in the text (section 5.2), although explanations are provided elsewhere in the manuscript. Still, it might be worth pointing out the occurrence of this other gradient switch in Fig. 10a and relating it with the gradient switch that occurs with the increasing discharge (appearing in the same figure).

**Our reply:** Many thanks for pointing this out. In the revised paper, we shall add '$10^4$' for the scale of $x$ axis (see Figure R5 below). In addition, in the main text, we will supplement the following sentences in section 5.2: "*In addition, it can be seen from Figure 10a that there exists a threshold of approximate 15000 m³/s for the tidal damping with respect to the position along the estuary. At lower river discharges (Q<15000 m³/s), the damping number δ tends to decrease (indicating a strengthening damping) in the landward direction, whereas it is the opposite at higher river discharges (Q<15000 m³/s).*"

[Figure]

Figure R5. Relationship between the tidal damping number $\delta$ (a), the residual water level slope $S$ (b), the residual water level $\bar{Z}$ (c) and the corresponding river discharge $Q$ imposed at the DT hydrological station for different positions, indicated by different symbols. The dashed lines with the same colour as the symbols were used to identify the critical river discharge for the maximum tidal damping (corresponding to the minimum value of $\delta$ in subplot a).

References:

Alebregtse, N. C. and de Swart, H. E.: Effect of river discharge and geometry on tides and net water transport in an estuarine network, an idealized model applied to the Yangtze Estuary, Cont. Shelf. Res., 123, 29–49, doi:10.1016/j.csr.2016.003.028, 2016.

Cai, H., Yang, Q., Zhang, Z., Guo, X., Liu, F., and Ou, S.: Impact of river-tide dynamics on the temporal mean water level profile in an estuary with substantial fresh water discharge, Hydrol. Earth Syst. Sci., 20, 1177–1195, doi:10.5194/hess-20-1177-2016, 2016.

Lu, S., Tong,C., Lee, D.Y., Zheng, J., Shen, J., Zhang, W., and Yan, Y.: Propagation of tidal waves up in Yangtze Estuary during the dry season, J. Geophys. Res., 120, 6445–6473, doi:10.1002/2014JC010414, 2015.

Hoitink, A. J. F., and Jay, D. A.: Tidal river dynamics: Implications for deltas. Reviews of Geophysics, 54, 240-272, doi:10.1002/2015rg000507, 2016.

Matte, P., Secretan, Y., & Morin, J.: Reconstruction of tidal discharges in the St. Lawrence fluvial estuary: The method of cubature revisited. J. Geophys. Res., 123, 5500-5524, doi:10.1029/2018JC013834, 2018.

Matte, P., Secretan, Y., and Morin, J.: Drivers of residual and tidal flow variability in the St. Lawrence fluvial estuary: Influence on tidal wave propagation. Continental Shelf Research, doi:10.1016/j.csr.2018.12.008, 2019.

Zhang, W., Feng, H. C., Hoitink, A. J. F., Zhu, Y. L., Gong, F.: Tidal impacts on the subtidal flow division at the main bifurcation in the Yangtze River Delta, Estuar. Coast Shelf S., 196, 301-314, doi:10.1016/j.ecss.2017.07.008, 2017.

Zhang, F., Sun, J., Lin, B., and Huang, G.: Seasonal hydrodynamic interactions between tidal waves and river flows in the Yangtze Estuary, J. Marine Syst., 186, 17–28, doi:10.1016/j.jmarsys.2018.05.005, 2018.

---

## Referee Comment (RC2) · Anonymous Referee #2 · 29 Mar 2019

General comments:

The manuscript examines the importance of river discharge on tidal damping, residual water level slopes and channel convergence in a seasonal scale in the Yangtze estuary. An analytical model for the tide-river dynamics has been used to understand the underlying mechanisms based on the previous works by the same authors and previous reports from spectra analysis of observed data by other researchers. The authors have identified a critical position of maximum tidal damping along the estuary for a given river discharge as wet or dry season. They also have identified a critical

value of river discharge at a given location, beyond which the tidal damping is reduced with increasing river discharge. It is contrary to the common assumption that larger river discharge leads to heavier tidal damping, which is driven by the cumulative effect of residual water level and channel convergence. This is the most important new insight of present manuscript to enhance our understanding of the nonlinear tide-river interactions and guide effective water management in the Yangtze estuary and other estuaries although the methods used were presented. The subject is relevant to the journal, the manuscript is well written and structured. The result analysis is thorough and the discussion is well presented. In conclusion, I recommend its publication after minor revision.

Specific comments:

L55-56: "little effort has been devoted to exploring the effect of river discharge on channel convergence, which is the other control factor for tide-river dynamics": How can the river discharge affect channel convergence? Provide explanations and references.

L105-106: "Datong hydrological station (where the tidal limit is)": As the authors have read reference about the fluctuation of tidal limit in the Yangtze estuary, you should note the significant fluctuation of the tidal limit during the similar period to the present manuscript. And one of the main identification result by the authors is the critical position of tidal damping controlled by the river discharge. Provide some explanations as the tidal limit is directly relevant to the effect of river discharge on the tidal damping and residual water level. In particular, suggest the authors to insert more words of relevant discussion into the section 5.

L231-234: "a threshold, corresponding to a critical value of river discharge, beyond which the relationship between the tidal damping rate and river discharge switches from negatively to positively correlated": Why the channel geometry is missing for the reason explanation of switch occurred here. Please insert more words into the section 5 of discussion about the correlation of critical value of river discharge with the channel

convergence.

Technical corrections:

L353: Replace St by Sr.

L357: Insert a blank space between S and a.

---

## Author Comment (AC2) · 31 Mar 2019

**Responses to comments by Reviewer #2**

We thank the Reviewer's careful consideration of our work. In this rebuttal, we have addressed all the comments formulated by the Reviewer by replying (in black) to her/his remarks (in blue).

General comments:
The manuscript examines the importance of river discharge on tidal damping, residual water level slopes and channel convergence in a seasonal scale in the Yangtze estuary. An analytical model for the tide-river dynamics has been used to understand the underlying mechanisms based on the previous works by the same authors and previous reports from spectra analysis of observed data by other researchers. The authors have identified a critical position of maximum tidal damping along the estuary for a given river discharge as wet or dry season. They also have identified a critical value of river discharge at a given location, beyond which the tidal damping is reduced with increasing river discharge. It is contrary to the common assumption that larger river discharge leads to heavier tidal damping, which is driven by the cumulative effect of residual water level and channel convergence. This is the most important new insight of present manuscript to enhance our understanding of the nonlinear tide-river interactions and guide effective water management in the Yangtze estuary and other estuaries although the methods used were presented. The subject is relevant to the journal, the manuscript is well written and structured. The result analysis is thorough and the discussion is well presented. In conclusion, I recommend its publication after minor revision.
**Our reply:** We thank the Reviewer for her/his overall positive assessment of our work.

Specific comments:
L55-56: "little effort has been devoted to exploring the effect of river discharge on channel convergence, which is the other control factor for tide-river dynamics": How can the river discharge affect channel convergence? Provide explanations and references.
**Our reply:** In the revised paper, we shall cite two recent publications concerning the impact of river discharge on channel convergence and revise the sentence into:

"*However, little effort has been devoted to exploring the effect of river discharge on channel convergence (represented by the gradient of cross-sectional area), which is the other control factor for tide-river dynamics (e.g., Matte et al., 2018, 2019). In particular, the river discharge affects the channel convergence primarily through*

*residual water level and hence water depth and cross-sectional area (Cai et al., 2014, 2016).*"

L105-106: "Datong hydrological station (where the tidal limit is)": As the authors have read reference about the fluctuation of tidal limit in the Yangtze estuary, you should note the significant fluctuation of the tidal limit during the similar period to the present manuscript. And one of the main identification result by the authors is the critical position of tidal damping controlled by the river discharge. Provide some explanations as the tidal limit is directly relevant to the effect of river discharge on the tidal damping and residual water level. In particular, suggest the authors to insert more words of relevant discussion into the section 5.

**Our reply:** We thank the reviewer for pointing this out. In the revised paper, we shall supplement the discussion part by including the following sentence:

"*For instance, Cai et al. (2019) explored how the freshwater regulation of the Three Gorges Dam (the world's largest hydroelectric station in terms of installed power capacity) may affect the alteration of tidal limit in the Yangtze estuary by means of the analytical model proposed in this paper. It was shown that the largest change of tidal limit by around 75 km occurred in October owing to the substantial increase in freshwater discharge.*" For more details with regard to the impact of freshwater discharge on the movement of tidal limit, readers can kindly refer to Cai et al. (2019).

L231-234: "a threshold, corresponding to a critical value of river discharge, beyond which the relationship between the tidal damping rate and river discharge switches from negatively to positively correlated": Why the channel geometry is missing for the reason explanation of switch occurred here. Please insert more words into the section 5 of discussion about the correlation of critical value of river discharge with the channel convergence.

**Our reply:** Indeed, here we did not provide detailed explanations with regard to the relationship between the switch of tidal damping and the channel geometry. This is because here (section 4.1) we aim to illustrate the phenomenon of maximum tidal damping based on the observed time series on a monthly scale. Hence, in the revised paper, we shall mention that "*The underlying mechanism will be elaborated further in the discussion part (see Section 5.2).*"

In particular, in section 5.2 we explicitly mentioned that:

"*The underlying mechanism for achieving a critical river discharge for maximum tidal damping can be primarily attributed to the cumulative effect of residual water level Z altering the water depth and hence the channel convergence and effective friction,*

*according to the definitions of estuary shape number γ and friction number χ in Table 1. Figure 11 presents these two controlling parameters (γ and χ) as a function of river discharge Q. It can be clearly seen in Figure 11a that there exists an apparent switch of the estuary shape number γ from positive (indicating a reduction of cross-sectional area in the landward direction) to negative (indicating an increase of cross-sectional area in the landward direction). In addition, more river discharge is required to achieve a switch in estuary shape number γ for the seaward positions where tide exerts more influence. The main reason for such a switch is the consistent increase of residual water level and hence water depth and cross-sectional area with river discharge.*"

Technical corrections:

L353: Replace St by Sr.

**Our reply:** Corrected as suggested.

L357: Insert a blank space between S and a.

**Our reply:** Corrected as suggested.

References:

Cai, H., Savenije, H. H. G., and Toffolon, M.: Linking the river to the estuary, influence of river discharge on tidal damping, Hydrol. Earth Syst. Sci., 18(1), 287-304, https://doi.org/10.5194/hess-18-287-2014, 2014.

Cai, H., Savenije, H. H. G., Jiang, C. Zhao L., Yang Q.: Analytical approach for determining the mean water level profile in an estuary with substantial fresh water discharge, Hydrol. Earth Syst. Sci., 20, 1-19, https://doi.org/10.5194/hess-20-1-2016, 2016.

Cai, H., Zhang, X., Guo, L., Zhang, M., Liu, F., and Yang, Q.: Impacts of Three Gorges Dam's operation on spatial-temporal patterns of tide-river dynamics in the Yangtze River estuary, China, Ocean Sci. Discuss., https://doi.org/10.5194/os-2018-138, in review, 2019.

Matte, P., Secretan, Y., Morin, J.: Reconstruction of tidal discharges in the St. Lawrence fluvial estuary: The method of cubature revisited. J. Geophys. Res., 123, 5500-5524, https://doi.org/10.1029/2018JC013834, 2018.

Matte, P., Secretan, Y., and Morin, J.: Drivers of residual and tidal flow variability in the St. Lawrence fluvial estuary: Influence on tidal wave propagation. Continental Shelf Research, https://doi.org/10.1016/j.csr.2018.12.008, 2019.

---

## Author Response (AR1)

**Response letter**

We thank the two Reviewers for the careful consideration of our work. Their constructive and thoughtful comments and suggestions led to a much improved and complete revision of the manuscript. In the revised paper, we have addressed all the comments formulated by the Reviewers by replying (in black) to their remarks (in blue). The line numbers in this rebuttal refer to the revised version of the manuscript.

**Responses to comments by Reviewer #1**

This manuscript investigates the influence of river discharge on tidal damping and residual water level slopes in the Yangtze River estuary at the seasonal scale. Building on previous work by the same author(s), an analytical model for river-tide dynamics is used to understand the underlying mechanisms responsible for the observed variability. Of particular interest, the authors identified (1) a critical value of river discharge, at a given location, beyond which tidal damping is reduced with increasing discharge, and (2) a critical position along the estuary, for a given discharge (e.g. wet or dry season), upstream of which tidal damping is reduced in the landward direction. Although the methods used were presented before, this application to a large estuary reveals new insights into the seasonal patterns of river-tide dynamics, which have implications for sustainable water management and sediment transport, as stressed by the authors. The subject is thus quite relevant; the manuscript is well written, well documented and clearly structured. The result analysis is thorough and a good discussion is presented. Overall, this is a very good paper. I recommend its publication after minor revision. My comments are detailed below.

**Our reply:** We thank the Reviewer for her/his overall positive assessment of our work.

General comments:
Do the critical values found in (1) and (2) (see above) represent the same phenomenon after all? It seems like, for a given (constant) discharge, when you move upstream a tidal river, the relative influence of river discharge increases, which is analogous to a

discharge increase at a given (fixed) location. Is this a good reasoning? If relevant, a word on the similarity/dissimilarity between the two processes could be said.

**Our reply:** We thank the Reviewer for this comment. Indeed, the underlying mechanism of generating critical position and river discharge is the same. In the revised paper, we have included the following sentence in the Conclusions part:

"*It is worth noting that the underlying mechanism of generating critical position along the estuary is similar to that of generating critical river discharge due to the fact that for a given (constant) river discharge, the more upstream in a tidal river, the stronger effect caused by the river discharge, which is analogous to a river discharge increase at a given (fixed) location.*" (Please see lines 477-481)

Discussion: A word on applicability/transferability of the method to other systems or other dynamical contexts should be added in the discussion. In particular, would this analytical approach work in systems with mixed diurnal/semidiurnal tides, in nonconvergent estuaries, or in estuaries with irregular (non-rectangular) cross-sections? Would it be possible to reduce the temporal averaging window to analyse the neap-spring variability in tidal damping and residual water level slopes? Similarly, could the method be adapted to rapidly varying flows? What adaptations would be necessary to include these aspects, if possible? I am not asking that the authors make those changes, but a discussion on limitations (and possible upgrades) of the proposed methodology would be useful.

**Our reply:** We agree with the Reviewer for this comment. In the revised paper, we have supplemented a new subsection in the Discussion part to highlight the limitation and transferability of the analytical model:

"*Although the current analytical model can well reproduce the first-order tide-river dynamics, it also has some limitations. The fundamental assumption is that the tidal wave can be described by a combination of a steady residual term (generated by the river discharge) and a time-dependent harmonic wave (introduced by the tidal flow). Thus, the proposed model can only capture the tidal asymmetry caused by tide-river interaction while it neglects the tidal asymmetry introduced by astronomical tides (e.g., nonlinear interactions among $K_1$, $O_1$ and $M_2$), overtides (e.g., $M_4$) and compound tides (e.g., MSf). Consequently, the proposed analytical method is preferably applied to tidal rivers with a predominant tidal constituent (e.g., $M_2$ or $K_1$).*

*It is assumed that both the tidally averaged cross-sectional area and channel width can be approximated by exponential functions following Equations (6)-(7). However, this is not a restrictive assumption since the model in principle can be applied to an arbitrary estuarine shape (i.e., bed elevation and channel width), as long as the variation of the cross-section is gradual. The proposed model can also be used to*

*quantify the spring-neap variability of the tide-river dynamics based on daily averaged tidal amplitude and river discharge conditions (see example in Cai et al., 2016). However, the model cannot be used to explore the tide-river dynamics within a tidal cycle since it is based on a tidally averaged scale. This means that it may not be applicable to the cases with rapidly varying river discharge.*" (Please see lines 417-434)

The Yangtze River estuary does not seem to have sharp morphological breaks, based on Fig. 4. However, in systems where they occur, a shift in the tidal-fluvial conditions may be observed near these breaks. In such a case, the location of the boundary between the tide-dominated and river-dominated reaches may be invariant to changes in river flow (Hoitink & Jay, 2016). In this situation, what should be expected to be the consequence on the position of maximum tidal damping and maximum residual water level slope along the estuary, under different discharge conditions?

**Our reply:** Thanks a lot for raising such an interesting case study, although the current analytical model is not applicable for such a case since it is assumed that the variation of the cross-section is gradual. In this case, it is likely to have a local maximum tidal damping at the boundary between the tide-dominated and river-dominated reaches owing to the sudden change of cross-section. However, it is possible to apply the analytical model proposed in this study to the upstream river-dominated region for given a suitable tidal forcing condition at the boundary (i.e., the downstream end of the river-dominated reach). Thus, the dynamics of position of maximum tidal damping and maximum residual water level slope along the estuary under a wide range of river discharge conditions would be the same as presented in this study. Further study in estuaries with sharp morphological breaks is needed in order to understand the underlying mechanism of tide-river dynamics.

Specific comments:
L55-56: "the effect of river discharge on channel convergence, which is the other control factor for tide-river dynamics": Can you provide a reference, or is this new knowledge? Also, it is not quite clear at this stage in the manuscript how discharge can affect channel convergence. Can you explain in a few words?

**Our reply:** In the revised paper, we have modified the sentence into:

"*However, little effort has been devoted to exploring the effect of river discharge on channel convergence (represented by the gradient of cross-sectional area), which is the other control factor for tide-river dynamics (e.g., Matte et al., 2018, 2019). In particular, the river discharge affects the channel convergence primarily through residual water level and hence water depth and cross-sectional area (Cai et al., 2014b, 2016).*" (Please see lines 56-60)

L81-90: "Recently, idealized (or analytical) models with a strongly simplified geometry and flow characteristics were applied [: : :]. [The model] can reasonably reproduce the first-order tide-river dynamics (only considering a predominant tidal constituent)": In terms of justification of the method, considering the limitations of the analytical model, can you explain the interest or benefit of using such a simplified model compared to full numerical models?

**Our reply:** In the revised paper, we have included the following sentences to clarify the benefit of using a simplified analytical model when compared with the full numerical model:

"*Although the tide-river dynamics in terms of elevation and velocity fields can be accurately simulated using fully nonlinear numerical models (e.g., Zhang et al., 2015a, b, 2018), the cause–effect relations (e.g. the impact of river discharge on tidal damping) cannot be explicitly identified by single realizations of numerical runs. To this aim, analytical models are valuable instruments that can provide a straightforward insight. Additionally, analytical models only need a minimum amount of data, and can explicitly provide estimates of integral quantities (e.g. tidal amplitude, velocity amplitude, wave celerity and phase lag), while numerical models need to reconstruct them from temporal and spatial time series.*" (Please see lines 85-92)

L92: "previous studies mainly focused on the tidal properties near the estuary mouth": Can you please provide references supporting this affirmation?

**Our reply:** In the revised paper, we have added the following three publications (Please see lines 104-105):

Alebregtse, N. C. and de Swart, H. E.: Effect of river discharge and geometry on tides and net water transport in an estuarine network, an idealized model applied to the Yangtze Estuary, Cont. Shelf. Res., 123, 29–49, https://doi.org/10.1016/j.csr.2016.003.028, 2016.

Lu, S., Tong,C., Lee, D.Y., Zheng, J., Shen, J., Zhang, W., and Yan, Y.: Propagation of tidal waves up in Yangtze Estuary during the dry season, J. Geophys. Res., 120, 6445–6473, https://doi.org/10.1002/2014JC010414, 2015.

Zhang, W., Feng, H. C., Hoitink, A. J. F., Zhu, Y. L., and Gong, F.: Tidal impacts on the subtidal flow division at the main bifurcation in the Yangtze River Delta, Estuar. Coast Shelf S., 196, 301-314, https://doi.org/10.1016/j.ecss.2017.07.008, 2017.

L123-124: "The tidal amplitude is determined by averaging the flood and ebb tidal amplitudes": What do you mean by "flood and ebb tidal amplitudes"? Please clarify.

**Our reply:** In the revised paper, we have clarified the definition of the tidal amplitude used in this paper:

"*The tidal amplitude is defined as a half of the tidal range either during the flood or the ebb period and we determined the mean value by averaging the tidal amplitudes during flood and ebb periods.*" (Please see lines 135-137)

L126: Consider adding "monthly averaged" before "tidal amplitude and water level" and replacing "tidal amplitude" by "tidal range", if appropriate.

**Our reply:** Thanks a lot for pointing this out! We have replaced "tidal amplitude and water level" with "monthly averaged tidal range and residual water level". (Please see line 139)

L232-235 and Fig. 3a: "there is a threshold, corresponding to a critical value of river discharge, beyond which the relationship between the tidal damping rate and river discharge switches from negatively to positively correlated": This does not show clearly in Fig. 3a, because of the straight regression lines. Can you illustrate the observed shifts with dotted lines maybe?

**Our reply:** In order to highlight the threshold of river discharge, we fitted the tidal damping rate using the quadratic equation. The revised figure 3 is presented below (Figure R1):

[Figure]

Figure R1. Scatterplot of tidal damping rate $\delta_H$ (a) and residual water level slope $S$ (b) for different reaches in the Yangtze River estuary as a function of river discharge

observed at the DT hydrological station. Subplot (a) also presents the quadratic regression lines, while subplot (b) presents the linear regression lines.

L255: Using two different friction coefficients K for the seaward and landward regions creates a break in the results. Is it real? If not, consider making a smoother transition of friction between the two regions.

**Our reply:** We very much appreciate reviewer's comment, which is indeed helpful to improve the performance of the analytical model to reproduce the main tide-river dynamics along the estuary. In the revised paper, we have explicitly mentioned that:

"*The calibrated value of K is 80 $m^{1/3}s^{-1}$ in the seaward region (x=0-32 km), whereas a smaller value of K=55 $m^{1/3}s^{-1}$ is used in the river dominated region (x=52-450 km). Meanwhile, in order to avoid discontinuous jump caused by the adoption of different friction coefficients, we adopted a friction coefficient of K=80-55 $m^{1/3}s^{-1}$ (indicating a linear reduction of the friction coefficient) over the transitional reach (x=32-52 km).*" (Please see lines 266-272)

L284-285 and Fig. S3: "the seasonal behaviour of the critical phase lag is relatively irregular": It looks regular to me in Fig. S3, but inversely correlated with Q. Please adjust the text accordingly.

**Our reply:** We thank the Reviewer for this comment. In the revised paper, we have removed this sentence.

L304-308: It should be said more explicitly that both the maximum value of residual water level slope Smax and its position along the estuary (Fig. 7) are correlated with river discharge.

**Our reply:** In the revised paper, we have explicitly mentioned that:

"*This indicates that both the maximum value of residual water slope $S_{max}$ (Figure 7b) and its position along the estuary (Figure 7a) are positively correlated with river discharge.*" (Please see lines 321-323)

L310-312 and Fig. S4: Can you briefly explain why the position of Sr,max is landward of the other two terms (St,max and Str,max)?

**Our reply:** In the revised paper, we have explicitly mentioned that:

"*In addition, we note that the position of the maximum riverine component $S_r$ is landward of the corresponding maximum values of the other two contributions ($S_t$ and $S_{tr}$), which is mainly due to the relatively larger residual frictional effect introduced by the riverine forcing.*" (Please see lines 327-329)

L326-327: The position of maximum tidal damping is almost coincident with the maximum (not minimum) values of the wave celerity and the minimum values of the velocity number. Please correct.

**Our reply:** You are right! In the revised paper, we have corrected this mistake:

"*In addition, the position of maximum tidal damping (corresponding to the minimum value of damping number δ, indicated by the dashed black line) is almost coincident with the maximum value of the celerity number λ and the minimum value of the velocity number μ.*" (Please see lines 342-345)

L327-328: "The slightly lagged responses [: : :] are due to nonlinear interaction between these main tide-river dynamics parameters": Are you able to provide a more detailed physical explanation for it?

**Our reply:** We thank the Reviewer for this comment. In the revised paper, we have supplemented the following sentence:

"*This also indicates the significantly nonlinear effect caused by estuary shape, bottom friction and river discharge as the tidal wave propagating upriver.*" (Please see lines 346-348)

L329-330: Replace "is directly followed by" by "directly follows from". Can you explain the correlation between the phase lag ε and the other variables and its role in tidal wave propagation (damping and celerity) based on your results?

**Our reply:** Many thanks for the correction. In the revised paper, we have provided more details regarding the relationship between the phase lag ε and the other variables:

"*As can be seen from Figures 8a-d, in general the phase lag ε is positively correlated with the damping number δ and the velocity number μ, while it is negatively correlated with the celerity number λ. Unlike tide-dominated estuaries with negligible residual water level, the key parameter that determines the nonlinear relationship between the phase lag ε and the other variables (δ, μ, λ) in tidal rivers lies in the water depth, which is controlled by the dynamics of residual water level.*" (Please see lines 349-354)

L353: Replace St by Sr.

**Our reply:** Corrected as suggested. (Please see line 377)

L361-362: Please specify whether the negative (positive) gradient indicates a strengthening (weakening) damping with respect to Q or to the landward position along the estuary, or both.

**Our reply:** Here, the negative (positive) gradient indicates a strengthening (weakening) damping with respect to river discharge Q. In the revised paper, we have explicitly mentioned this point. (Please see lines 384-387)

L434-435: "this is the first study that shows the gradient switch of the cross-sectional area and tidal damping with the river discharge": This gradient switch in tidal damping was also recently documented by Matte et al., (2019) in the St. Lawrence River at the neap-spring and seasonal scale. I suggest referencing their work, here or elsewhere in the manuscript.

**Our reply:** We agree with your comment. In the revised paper, we have included two recent publications by Matte et al. (2018, 2019). (Please see lines 492-493)

L451-456: There is a duplication of references: Cai et al. (2012a) and Cai et al. (2012b) are the same.

**Our reply:** We have corrected this mistake in the revised paper.

Fig. 1: Can you add river kilometers at each station in panel (b)?

**Our reply:** Yes! In the revised paper, we have added river kilometers at each station in panel (b). The updated figure is presented below (Figure R2).

[Figure]

Figure R2. Sketch map of the Yangtze River basin (a) and the Yangtze River estuary (b) displaying the location of gauging (triangle) and hydrological (circle) stations.

Figs. 2 and 8: I find it hard to differentiate the pink, red and/or dark red curves. Can you use more contrasting colors?

**Our reply:** In the revised paper, we have modified these Figures (see Figures R3 and R4 below).

[Figure]

Figure R3. Temporal (monthly averaged) variations of observed tidal range $H$ (a) and residual water level $\bar{Z}$ (b) at different gauging stations along the Yangtze River estuary together with the observed river discharge at Datong hydrological station.

[Figure]

Figure R4. Longitudinal variation of the main tide-river dynamics (a, b, c, d) and contributions of tidal and riverine forcing to the residual water level slope together with the water depth (e, f) for the wet (a, c, e) and dry seasons (b, d, f) in the Yangtze estuary. The dashed lines in each subplot represent the critical position for maximum tidal damping (corresponding to the minimum value of damping number $\delta$).

Fig. 10: Add "x 10^4" to the scale for river discharge Q. In Fig. 10a, there is another gradient switch happening around 15 000 m3/s with respect to the position along the estuary. At lower discharges, maximum damping occurs seaward, whereas at higher discharges, it occurs landward. This was not described in the text (section 5.2), although explanations are provided elsewhere in the manuscript. Still, it might be worth pointing out the occurrence of this other gradient switch in Fig. 10a and relating it with the gradient switch that occurs with the increasing discharge (appearing in the same figure).

**Our reply:** Many thanks for pointing this out. In the revised paper, we have added '$10^4$' for the scale of $x$ axis (see Figure R5 below). In addition, in the main text, we have supplemented the following sentences in section 5.2:

"*In addition, it can be seen from Figure 10a that there exists a threshold of approximate 15000 m³/s for the tidal damping with respect to the position along the estuary. At lower*

*river discharges (Q<15000 m³/s), the damping number δ tends to decrease (indicating a strengthening damping) in the landward direction, whereas it is the opposite at higher river discharges (Q>15000 m³/s).*" (Please see lines 394-398)

[Figure]

Figure R5. Relationship between the tidal damping number $\delta$ (a), the residual water level slope $S$ (b), the residual water level $\bar{Z}$ (c) and the corresponding river discharge $Q$ imposed at the DT hydrological station for different positions, indicated by different symbols. The dashed lines with the same colour as the symbols were used to identify the critical river discharge for the maximum tidal damping (corresponding to the minimum value of $\delta$ in subplot a).

**Responses to comments by Reviewer #2**

General comments:
The manuscript examines the importance of river discharge on tidal damping, residual water level slopes and channel convergence in a seasonal scale in the Yangtze estuary. An analytical model for the tide-river dynamics has been used to understand the underlying mechanisms based on the previous works by the same authors and previous reports from spectra analysis of observed data by other researchers. The authors have

identified a critical position of maximum tidal damping along the estuary for a given river discharge as wet or dry season. They also have identified a critical value of river discharge at a given location, beyond which the tidal damping is reduced with increasing river discharge. It is contrary to the common assumption that larger river discharge leads to heavier tidal damping, which is driven by the cumulative effect of residual water level and channel convergence. This is the most important new insight of present manuscript to enhance our understanding of the nonlinear tide-river interactions and guide effective water management in the Yangtze estuary and other estuaries although the methods used were presented. The subject is relevant to the journal, the manuscript is well written and structured. The result analysis is thorough and the discussion is well presented. In conclusion, I recommend its publication after minor revision.

**Our reply:** We thank the Reviewer for her/his overall positive assessment of our work.

Specific comments:

L55-56: "little effort has been devoted to exploring the effect of river discharge on channel convergence, which is the other control factor for tide-river dynamics": How can the river discharge affect channel convergence? Provide explanations and references.

**Our reply:** In the revised paper, we have cited two recent publications concerning the impact of river discharge on channel convergence and revised the sentence into:

"*However, little effort has been devoted to exploring the effect of river discharge on channel convergence (represented by the gradient of cross-sectional area), which is the other control factor for tide-river dynamics (e.g., Matte et al., 2018, 2019). In particular, the river discharge affects the channel convergence primarily through residual water level and hence water depth and cross-sectional area (Cai et al., 2014b, 2016).*" (Please see lines 56-69)

L105-106: "Datong hydrological station (where the tidal limit is)": As the authors have read reference about the fluctuation of tidal limit in the Yangtze estuary, you should note the significant fluctuation of the tidal limit during the similar period to the present manuscript. And one of the main identification result by the authors is the critical position of tidal damping controlled by the river discharge. Provide some explanations as the tidal limit is directly relevant to the effect of river discharge on the tidal damping and residual water level. In particular, suggest the authors to insert more words of relevant discussion into the section 5.

**Our reply:** We thank the reviewer for pointing this out. In the revised paper, we have supplemented the discussion part by including the following sentence:

"*For instance, Cai et al. (2019) explored how the freshwater regulation of the Three Gorges Dam (the world's largest hydroelectric station in terms of installed power capacity) may affect the alteration of tidal limit in the Yangtze estuary by means of the analytical model proposed in this paper. It was shown that the largest change of tidal limit by around 75 km occurred in October owing to the substantial increase in freshwater discharge.*" For more details with regard to the impact of freshwater discharge on the movement of tidal limit, readers can kindly refer to Cai et al. (2019). (Please see lines 443-447)

L231-234: "a threshold, corresponding to a critical value of river discharge, beyond which the relationship between the tidal damping rate and river discharge switches from negatively to positively correlated": Why the channel geometry is missing for the reason explanation of switch occurred here. Please insert more words into the section 5 of discussion about the correlation of critical value of river discharge with the channel convergence.

**Our reply:** Indeed, here we did not provide detailed explanations with regard to the relationship between the switch of tidal damping and the channel geometry. This is because here (section 4.1) we aim to illustrate the phenomenon of maximum tidal damping based on the observed time series on a monthly scale. Hence, in the revised paper, we have explicitly mentioned that "*The underlying mechanism will be elaborated further in the discussion part (see Section 5.2).*"

In particular, in section 5.2 we explicitly mentioned that:

"*The underlying mechanism for achieving a critical river discharge for maximum tidal damping can be primarily attributed to the cumulative effect of residual water level Z altering the water depth and hence the channel convergence and effective friction, according to the definitions of estuary shape number γ and friction number χ in Table 1. Figure 11 presents these two controlling parameters (γ and χ) as a function of river discharge Q. It can be clearly seen in Figure 11a that there exists an apparent switch of the estuary shape number γ from positive (indicating a reduction of cross-sectional area in the landward direction) to negative (indicating an increase of cross-sectional area in the landward direction). In addition, more river discharge is required to achieve a switch in estuary shape number γ for the seaward positions where tide exerts more influence. The main reason for such a switch is the consistent increase of residual water level and hence water depth and cross-sectional area with river discharge.*" (Please see lines 399-409)

Technical corrections:

**Our reply:** Corrected as suggested. (Please see line 377)

**Our reply:** Corrected as suggested. (Please see line 381)

[revised manuscript text omitted]